# Time-Resolved Visualization of Cyanotoxin Synthesis via Labeling by the Click Reaction in the Bloom-Forming Cyanobacteria *Microcystis aeruginosa* and *Planktothrix agardhii*

**DOI:** 10.3390/toxins17060278

**Published:** 2025-06-03

**Authors:** Rainer Kurmayer, Rubén Morón Asensio

**Affiliations:** 1Research Department for Limnology, University of Innsbruck, Mondseestrasse 9, 5310 Mondsee, Austria; ruben.moron-asensio@student.uibk.ac.at; 2Forschungszentrum Ökologie des Alpinen Raumes, Universität Innsbruck, Innrain 52, 6020 Innsbruck, Austria

**Keywords:** microcystin, anabaenopeptin, time-lapse experiments, pulse-feeding, chemo-selective labeling, high-resolution microscopy, flow cytometry

## Abstract

In non-ribosomal peptide synthesis of cyanobacteria, promiscuous adenylation domains allow the incorporation of clickable non-natural amino acids into peptide products—namely into microcystins (MCs) or into anabaenopeptins (APs): 4-azidophenylalanine (Phe-Az), *N*-propargyloxy-carbonyl-L-lysine (Prop-Lys), or *O*-propargyl-L-tyrosine (Prop-Tyr). Subsequently, chemo-selective labeling is used to visualize the clickable cyanopeptides using Alexa Fluor 488 (A488). In this study, the time-lapse build up or decline of azide- or alkyne-modified MCs or APs was visualized during maximum growth, specifically MC biosynthesis in *Microcystis aeruginosa* and AP biosynthesis in *Planktothrix agardhii*. Throughout the time-lapse build up or decline, the A488 signal occurred with heterogeneous intracellular distribution. There was a fast increase or decrease in the A488 signal for either Prop-Tyr or Prop-Lys, while a delayed or unobservable A488 signal for Phe-Az was related to increased cell size as well as a reduction in growth and autofluorescence. The proportion of clickable MC/AP in peptide extracts as recorded by a chemical–analytical technique correlated positively with A488 labeling intensity quantified via laser-scanning confocal microscopy for individual cells or via flow cytometry at the population level. It is concluded that chemical modification of MC/AP can be used to track intracellular dynamics in biosynthesis using both analytical chemistry and high-resolution imaging.

## 1. Introduction

Cyanobacteria frequently dominate aquatic ecosystems and have received much attention due to their ability to produce numerous toxic secondary metabolites, so-called cyanotoxins, for example, microcystins (MCs), anatoxins, and cylindrospermopsins. Cyanotoxin production is a global problem as it negatively affects freshwater quality. It is generally agreed that cyanobacterial harmful algal blooms (cHABs) are increasing in frequency, a phenomenon that is to some extent attributable to global warming and eutrophication [1,2]. The negative ecological and health impacts of cHABs have been documented, although there are still significant gaps in our understanding of cyanotoxin function. One remaining challenge is to understand the synthesis and regulation of cyanotoxins in the cell, which is also linked to the function of these toxins in nature.

Aside from ecosystem relevance, cyanotoxins comprise pharmacologically active compounds with potential antibacterial, fungal, or cytostatic effects. MCs, as well as many other toxic/bioactive metabolites, are synthesized by large multifunctional enzyme complexes via the thio-template mechanism, known as non-ribosomal peptide synthesis (NRPS) [3]. Through this pathway, both proteinogenic and non-proteinogenic amino acids (AAs) are sequentially incorporated into the growing peptide chain [4]. For example, it has been estimated that 48 steps are needed for the biosynthesis of MCs, of which 45 have been assigned to catalytic NRPS domains [5].

Common bloom-forming cyanobacteria include the genera *Dolichospermum*, *Microcystis*, and *Planktothrix*, which produce MCs and/or anabaenopeptins (APs) among other non-ribosomally synthesized peptides, such as aeruginosins, cyanopeptolins, and microginins. Moreover, they synthesize ribosomally and post-translationally modified peptides such as aerucyclamides, microviridins, and prenylagaramides [6]. For MCs, inhibition of protein phosphatase 1 and 2A in the nanomolar concentration range has been observed repeatedly [1,2,3]. For other peptides, such as APs, inhibition of serine proteases and the resulting toxicity to aquatic crustaceans has been reported [6,7,8]. It has been postulated that intercellular cocktails of various toxins or peptides may provide better protection due to synergistic interactions but also make co-evolutionary responses more difficult [9]. This hypothesis is supported by the observation that, for certain peptides synthesized via NRPS, there is a trend in evolution toward increased structural diversity, either via domain duplication [10] or point mutation and recombination within NRPS domains [11,12].

The same genotypes produce MC structural variants with variable AAs—that is, MC-RR vs. MC-LR and MC-HtyR—implying co-production of MC variants carrying Arg or Leu and Hty in position 2 of the MC-molecule [13,14,15]. Similarly, research has documented the co-occurrence of AP B or AP F and AP A or OscY [11,12,16], implying co-production of AP variants carrying Arg and Tyr in exocyclic position 1 of the AP molecule. For AP synthetase, we have genetically characterized and heterologously expressed the adenylation (A-)domain of the initiation module (ApnAA1) of the NRPS [11]. It was used in vitro as a model system to biochemically characterize the genetic variation observed within these A-domains. This endeavor provided the first evidence that particular A-domain genotypes are capable of activation of chemically divergent AAs—that is, Arg vs. Tyr—with comparable efficiency. The findings also demonstrated promiscuity for other proteinogenic AAs. In another study, the authors crystallized such an A-domain and elucidated the structural basis of substrate activation; they described a so-called bispecific A-domain [17]. From this crystallographic analysis, it could be inferred that ApnAA1-domain genotypes carrying substitutions defining the so-called specificity conferring code [18] led to a conformational change that allows for Arg and Tyr activation in parallel. Thus, only a few critical substitutions that cause a conformational change during substrate activation are responsible for the observed promiscuity. Subsequently, research has identified more genotypes from strains co-producing APs with three or more different AAs (Arg or Tyr or Lys) in position 1 [12].

Understanding the molecular basis for such promiscuity has opened the possibility of introducing unnatural AAs (non-AAs) with azide or alkyne groups into MCs or APs, enabling visualization via a subsequent click chemistry reaction [17,19]. In general, click chemistry involves the introduction of a functional group that is covalently attached to a substrate and introduced into a biomolecule of interest (i.e., introducing a non-AA into a peptide). The functional group reacts chemo-selectively, in the presence of all the functional groups found in a living cell, with an added chemical probe (i.e., Alexa Fluor 488 [A488]), through a chemical reaction such as the classical copper-catalyzed azide-alkyne cycloaddition (CuAAC) [20,21]. For this aim, non-AAs have been tested (i.e., 4-azidophenylalanine [Phe-Az], *N*-propargyl-lysine [Prop-Lys], and H-L-propargyl-tyrosine [Prop-Tyr]) to enable subsequent biomolecular labeling of the resulting modified peptide using A488 fluorophore azide or alkyne [21,22,23] (Figure 1).

Although MC/AP fluorescent labeling by chemo-selective reaction has been documented [21,22], questions on the labeling sensitivity and background fluorescence through autofluorescence (AF) remain. Surface bloom–forming cyanobacteria, such as *Microcystis aeruginosa,* have lower photosynthetic efficiency than cyanobacteria stratifying in the metalimnion, such as *Planktothrix* spp. (e.g., see Figures 5.4 and 5.5 in [24]). For example, *M. aeruginosa* has a lower chlorophyll *a* specific absorption efficiency compared with *Phormidium* and *Trichodesmium* and other taxa of the Phormidiaceae (Microcoleaceae) sensu [25,26]. In contrast, *Planktothrix* spp. has higher phycobilin contents, particularly phycoerythrin [27], enabling relatively higher growth rate under shade conditions [28].

The higher AF in *Planktothrix agardhii* may also lead to an increased signal-to-noise ratio. We recently tested a second fluorophore, Alexa Fluor 405 (A405), in the blue light spectrum, in parallel to A488 for *P. agardhii* and *M. aeruginosa* [22]. A405 also resulted in peptide labeling, but the labeling intensity was generally lower and the distinction from natural background AF was reduced, limiting its use in *Planktothrix* spp. Hence, the selection of the excitation/emission spectrum of the fluorophore is of particular significance for cyanobacteria with stronger AF, such as *Planktothrix* spp. Currently, it is not known how labeling signal intensity quantitatively relates to the clickable MC/AP content.

Time-lapse build up and decline experiments have been performed to better understand the synthesis of MC/AP through tracing via clickable non-AA incorporation [23]. The build up of A488 labeling signal in cells was observed by growing them in the presence of non-AAs under maximum growth conditions. Samples were taken every 12 h for the first 48 h, and a final sample was taken after 168 h (7 days). In addition, the decline of labeling signal was also determined by growing the cells under maximum growth conditions. At T0, following a 48 h incubation with non-AAs, the cells were transferred to fresh medium and harvested directly after washing and sampled every 24 h. In the present study, the results for A488 signal intensity were compared directly to the clickable AP/MC content as revealed by chemical–analytical methods reported earlier [23]. The A488 signal intensity was determined using two independent methods: direct examination of individual cells with high-resolution microscopy and examination of the total population with flow cytometry (FCM). This study reports a quantitative correlation between the amount of clickable MC/AP determined through chemical analysis and A488 signal intensity through the click chemical reaction.

**Figure 1 toxins-17-00278-f001:**
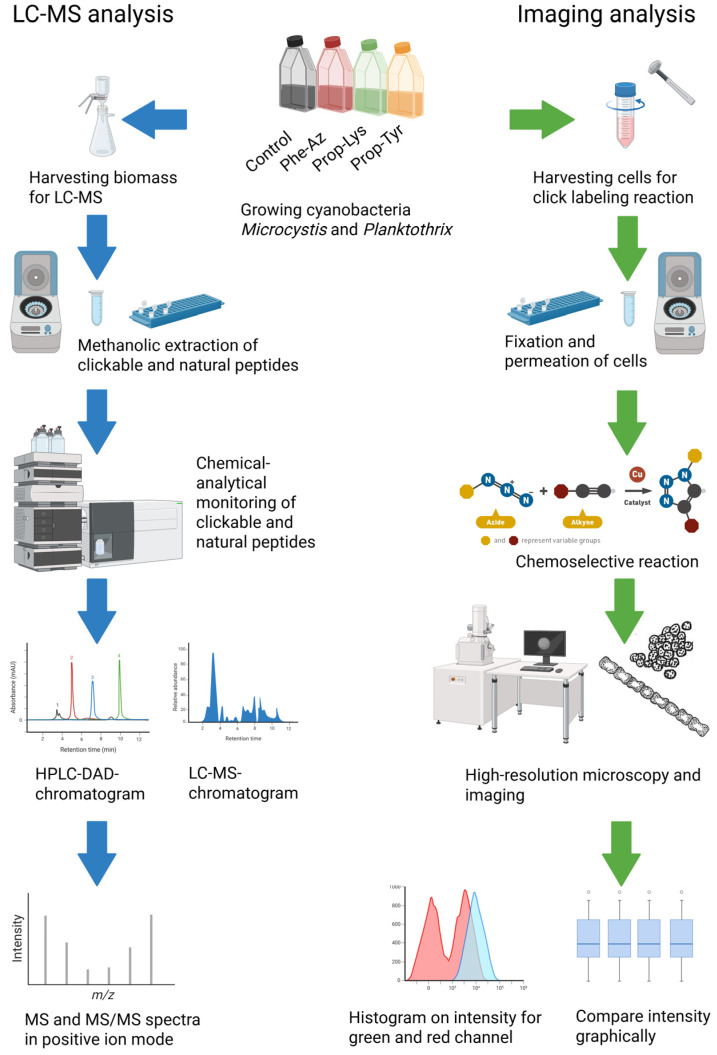
General overview of the protocol steps for the two tasks related to the click-chemical labeling of cyano-peptide synthesis in the cyanobacteria *Microcystis aeruginosa* or *Planktothrix agardhii*. The blue arrows indicate the protocol steps in the chemical-analytical analysis workflow [21,22,23], while the green arrows indicate the steps in chemo-selective labeling and microscopic analysis. Created in BioRender. Deng, L. (2025) https://BioRender.com/f6m1hhc.

## 2. Results

### 2.1. Qualitative Observation of A488 Signal Generation During Time-Lapse Build up via High-Resolution Microscopy

The A488 signal intensity was monitored qualitatively and quantitatively via laser-scanning confocal microscopy during time-lapse experiments for the cyanobacteria *M. aeruginosa* and *P. agardhii*. For *M. aeruginosa*, A488 signal build up resulting from clickable MC synthesis occurred quickly (within 12–24 h) for cells fed Prop-Tyr or Prop-Lys, while control cells (i.e., grown in the absence of non-AAs and processed under identical conditions) showed negligible A488 signal. For Phe-Az-fed cells, A488 signal build up was delayed, occurring after 24–36 h (Figure 2). Notably, the same heterogeneous distribution for the A488 signal reported in a previous study [21] was observed from the first time point after pulse feeding (i.e., at T1 [12 h]). In contrast, AF appeared to be distributed more homogeneously within the cell during the entire experimental period. For *P. agardhii*, A488 signal build up resulting from clickable AP synthesis also occurred quickly for cells fed Prop-Tyr or Prop-Lys (12–24 h). In contrast, there was no signal build up for Phe-Az-fed cells when compared with control cells (Figure 3). The delayed or unobservable A488 signal generation for Phe-Az-fed cells was related to the general reduction in growth, as reported previously [23]. Accordingly, for the Phe-Az-fed cells, the cell diameter and volume increased in both cyanobacteria compared with the control cells or Prop-Tyr- or Prop-Lys-fed cells (Table 1 and Appendix A). There was a rather heterogeneous distribution of A488 signal for the Prop-Tyr -and Prop-Tyr-fed cells from the beginning (T1 = 12 h), which will be investigated in detail in a forthcoming article. Similarly to *M. aeruginosa*, AF of *P. agardhii* was found to be distributed more evenly within the cells throughout the entire experiment.

### 2.2. Qualitative Observation of A488 Signal Reduction During Time-Lapse Decline via High-Resolution Microscopy

In addition to the increase in the A488 signal intensity, its decline was monitored during time-lapse experiments. Specifically, *M. aeruginosa* and *P. agardhii* were grown in the presence of non-AAs for 48 h and subsequently transferred to the new medium under maximum growth conditions. Similarly to the A488 signal intensity build up, the A488 signal intensity decline was rather fast: within 24 h for *M. aeruginosa* fed Prop-Tyr or Prop-Lys (Figure 4) and within 48 h for *P. agardhii* fed Prop-Tyr or Prop-Lys (Figure 5). There was a slower decline for Phe-Az-fed *M. aeruginosa*, while *P. agardhii* fed Phe-Az showed no difference compared with the control cells. Corresponding to the reduced growth rate, the Phe-Az fed cells also had an increased cell diameter and increased cell volume compared with the control cells or the Prop-Tyr or Prop-Lys fed cells (Table 1 and Appendix A).

In general, in *M. aeruginosa* the heterogeneous distribution of the A488 signal remained visible until the end of the experiment. In contrast, for *P. agardhii,* there was a temporal sequence leading to the disappearance of heterogeneous structures in the cell: Heterogeneous structures that occurred as distinct entities were replaced gradually and partially by ring-like formations in the peripheral area of the cell. The formation of these structures will be explored in more detail in a subsequent article.

### 2.3. Quantification of A488 Signal Intensity and AF via High-Resolution Microscopy During Time-Lapse Build up and Decline

Next, the labeling signal intensity was quantified by recording both the A488 signal intensity and AF per cell volume or per cell from 20 randomly selected cells per technical replicate and time point. Corresponding to the analysis of micrographs (Figure 2), A488 signal intensity increased quantitatively by 4.4- and 4.2-fold within 12 h for Prop-Tyr- and Prop-Lys-fed *M. aeruginosa*. In comparison, Phe-Az-fed *M. aeruginosa* showed a slower increase in labeling signal: 2.8-fold after 24 h, and a plateau phase after 36 h (Figure 6a and Appendix A).

The A488 signal intensity increased by 3.5- and 2.4-fold within 12 h for Prop-Lys- and Prop-Tyr-fed *P. agardhii*, respectively. In contrast, the A488 signal intensity from Phe-Az-fed cells remained low and did not differ from the control cells during the entire time-lapse signal build up experiment (Figure 6b). For both cyanobacteria, the Phe-Az-fed cells consistently showed the lowest AF per cell volume or cell (Figure 7 and Appendix A). Thus, Phe-Az feeding reduced cell division and growth [23], increased cell size, and significantly reduced AF during the entire experiment (Table 2 and Appendix A). Somewhat unexpectedly, the ratio of A488 intensity to AF intensity was high for Phe-Az-fed *M. aeruginosa* and even *P. agardhii* (Appendix A).

During the time-lapse signal decline experiment, both cyanobacteria showed a fast (24 h) decrease in A488 labeling intensity—by 4.4- and 3.2-fold for Prop-Tyr- and Prop-Lys-fed cells, respectively—while AF remained rather constant (Figure 6c,d vs. Figure 7c,d and Appendix A). Only Phe-Az-fed *M. aeruginosa* showed a delayed decline in A488 signal intensity (i.e., 2-fold at 48 h), which was attributed to the reduced growth rate. Accordingly, AF became significantly reduced in Phe-Az-fed cells during the entire experiment. As apparent in the micrographs (Figure 4 and Figure 5), Phe-Az-fed *P. agardhii* did not differ from the control cells at any time during the entire time-lapse decline experiment.

Signal co-localization analysis was used to quantify the heterogeneous distribution of A488. Specifically, the correlation of the voxels (three-dimensional pixels) between the A488 signal and the AF per cell was determined. In general, for *M. aeruginosa*, there was less difference in the co-localization indices between the treatments (Table 3 and Appendix A). In contrast, Prop-Lys- and Prop-Tyr-fed *P. agardhii* showed a significantly reduced correlation compared with the control cells. Note that, even when excluding Phe-Az-fed cells because of toxic side effects, the other two treatments differed significantly from the control cells. In summary, qualitative description of high-resolution micrographs was supported by quantitative evaluation of the A488 signal and AF.

**Figure 2 toxins-17-00278-f002:**
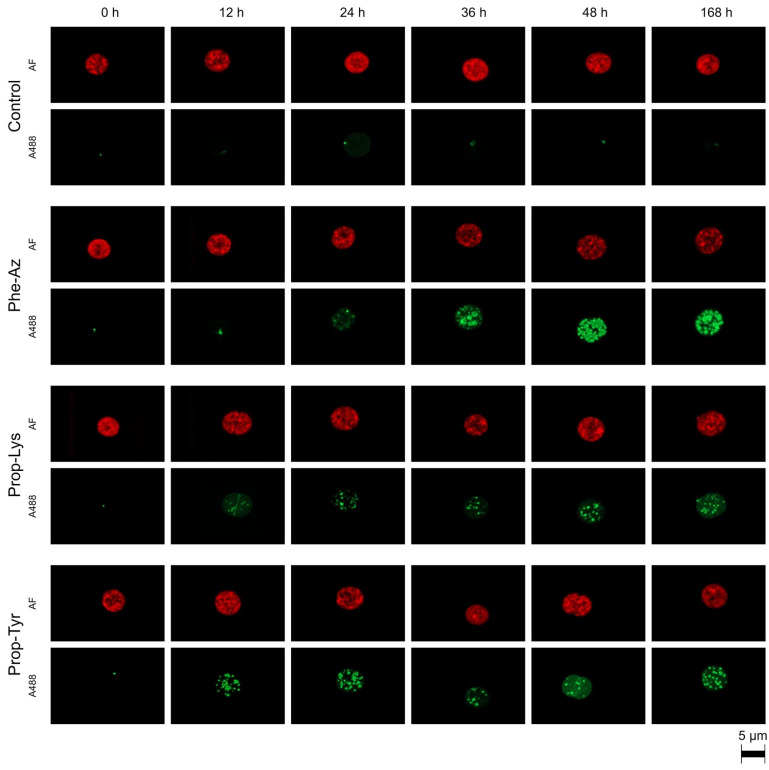
High-resolution micrographs of *M. aeruginosa* strain Hofbauer showing the increase in Alexa Fluor 488 (A488) labeling signal related to clickable MC synthesis via time-lapse build up during feeding of non-AAs (4-azidophenylalanine [Phe-Az], *N*-propargyl-lysine [Prop-Lys], and H-L-propargyl-tyrosine [Prop-Tyr]). Control cells were grown in the absence of non-AAs and processed under identical conditions. The natural autofluorescence (AF) is shown in red, while the A488 labeling signal is shown in green.

**Figure 3 toxins-17-00278-f003:**
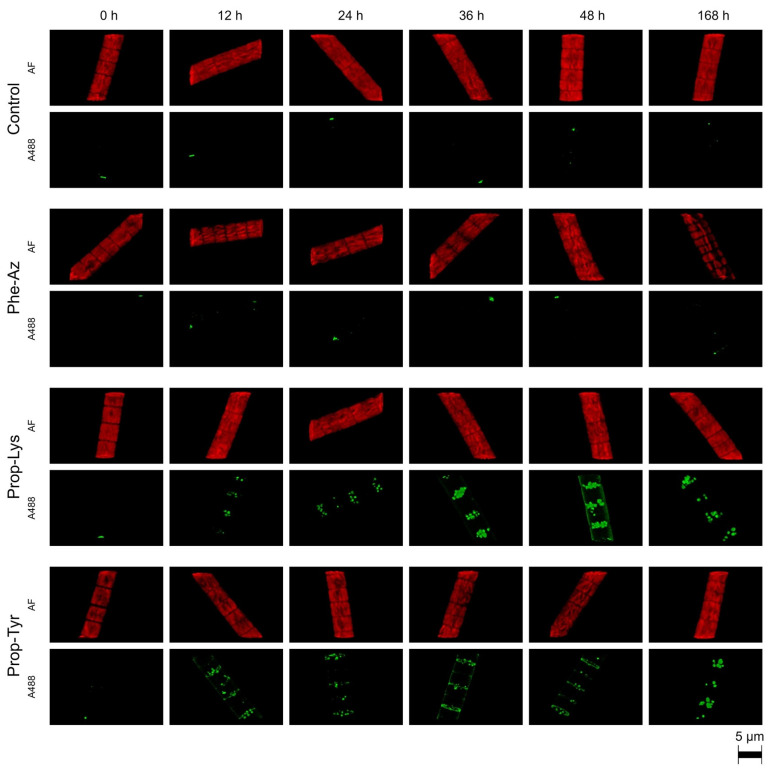
High-resolution micrographs of *P. agardhii* strain no371/1 showing the increase in Alexa Fluor 488 (A488) labeling signal related to clickable AP synthesis via time-lapse build up during feeding of non-AAs (Phe-Az, Prop-Lys, Prop-Tyr). Phe-Az-modified AP synthesis was not related to A488 labeling compared with control cells. The natural autofluorescence (AF) is shown in red, while the A488 labeling signal is shown in green.

**Figure 4 toxins-17-00278-f004:**
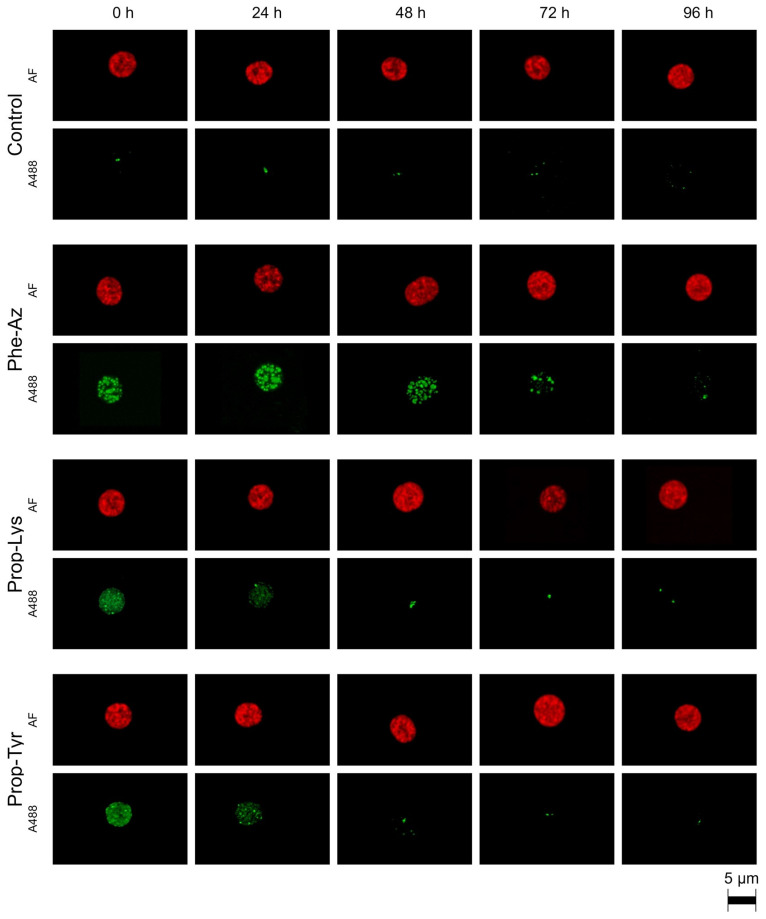
High-resolution micrographs of *M. aeruginosa* showing the decrease in Alexa Fluor 488 (A488) labeling related to the decline in clickable MC synthesis via pulsed feeding of non-AAs (Phe-Az, Prop-Lys, Prop-Tyr). The natural autofluorescence (AF) is shown in red, while the A488 labeling signal is shown in green.

**Figure 5 toxins-17-00278-f005:**
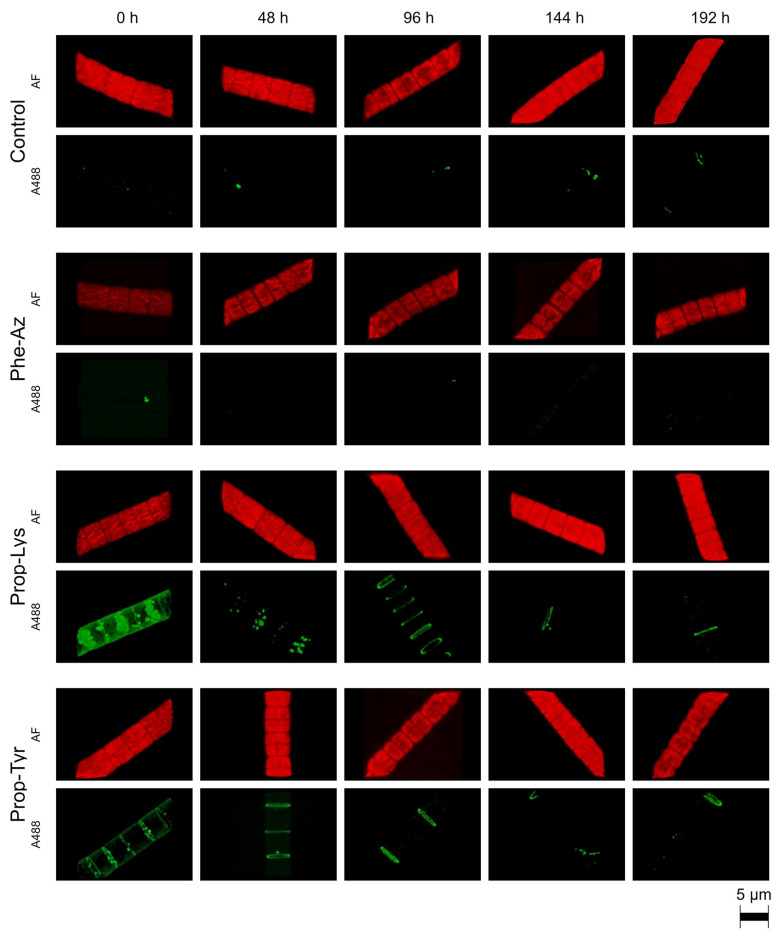
High-resolution micrographs of *P. agardhii* showing the decrease in A488 labeling signal related to the decline in clickable AP synthesis via pulsed feeding of non-AAs (Phe-Az, Prop-Lys, Prop-Tyr). Phe-Az-modified AP synthesis was not related to A488 labeling compared with control cells. The natural autofluorescence (AF) is shown in red, while the A488 labeling signal is shown in green.

**Figure 6 toxins-17-00278-f006:**
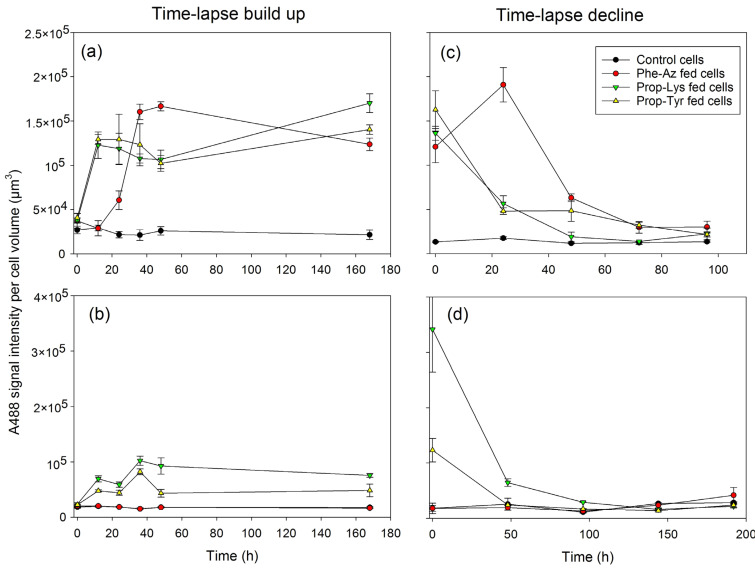
The mean (±SE) A488 signal intensities per cell volume (in µm^3^) of *M aeruginosa* (**a**,**c**) or *P. agardhii* (**b**,**d**) during time-lapse experiments related to the build up (**a**,**b**) or decline (**c**,**d**) of clickable MC (**a**,**c**) or clickable AP (**b**,**d**). A488 signal intensities were quantified from images as shown in Figure 2, Figure 3, Figure 4 and Figure 5.

**Table 2 toxins-17-00278-t002:** The mean (±SE) A488 signal intensity (×10^4^) and autofluorescence (AF) (×10^5^) per cell volume (in µm^3^) and A488/AF ratio during time-lapse signal build up and decline in *M. aeruginosa* and *P. agardhii*. The mean (±SE) fold changes compared with the control cells are given in parentheses. Graphical data are shown in Figure 6, Figure 7 and Appendix A. Measurements for each time point are shown in Appendix A.

Strain	Experiment	Control	Phe-Az	Prop-Lys	Prop-Tyr	*n*	*p*-Value ^1,2^ (Treatment)	*p*-Value ^3^ (Time)
A488 signal intensity (×10^4^)								
*M. aeruginosa*	build up	2.4 ± 0.18 ^a^	9.6 ± 1.4 ^b^ (4.2 ± 0.6)	11.1 ± 1 ^bc^ (4.7 ± 0.5)	11.1 ± 1 ^bc^ (4.7 ± 0.5)	72	<0.001 ***	<0.001 ***
	decline	1.4 ± 0.07 ^a^	8.7 ± 1.7 ^b^ (6.0 ± 1)	5 ± 1.1 ^c^ (3.6 ± 0.9)	6.3 ± 1.3 ^d^ (4.6 ± 1.1)	60	<0.001 ***	<0.001 ***
*P. agardhii*	build up	1.8 ± 0.07 ^a^	1.8 ± 0.07 ^a^ (1.0 ± 0.03)	7 ± 0.67 ^b^ (4.0 ± 0.4)	4.8 ± 0.48 ^ab^ (2.8 ± 0.3)	72	<0.001 ***	0.002 **
	decline	2.2 ± 0.25 ^a^	2.3 ± 0.4 ^b^ (1.1 ± 0.2)	9.5 ± 3.3 ^a^ (5.1 ± 2)	4.0 ± 1.2 ^a^ (2.1 ± 0.66)	60	0.003 **	0.002 **
AF intensity (×10^5^)								
*M. aeruginosa*	build up	0.8 ± 0.03 ^a^	0.6 ± 0.03 ^b^ (0.8 ± 0.04)	0.78 ± 0.03 ^a^ (1.0 ± 0.03)	0.76 ± 0.03 ^a^ (1.0 ± 0.03)	72	<0.001 ***	0.001 ***
	decline	3.5 ± 0.24 ^a^	2.4 ± 0.23 ^b^ (0.67 ± 0.06)	3.5 ± 0.2 ^a^ (1.0 ± 0.3)	3.5 ± 0.22 ^a^ (1.0 ± 0.3)	60	0.003 **	0.234
*P. agardhii*	build up	1.7 ± 0.09 ^a^	1.2 ± 0.08 ^b^ (0.8 ± 0.06)	1.5 ± 0.04 ^a^ (0.9 ± 0.03)	1.5 ± 0.06 ^a^ (0.9 ± 0.04)	72	0.003 **	0.003 **
	decline	6.1 ± 0.25 ^a^	1.9 ± 0.44 ^b^ (0.3 ± 0.08)	5.5 ± 0.21 (0.9 ± 0.03)	5.3 ± 0.22 ^a^ (0.9 ± 0.05)	60	<0.001 ***	0.011 *
A488/AF ratio								
*M. aeruginosa*	build up	0.32 ± 0.02 ^a^	1.98 ± 0.34 ^b^	1.48 ± 0.15 ^bc^	1.57 ± 0.16 ^bc^	72	<0.001 ***	<0.001 **
	decline	0.05 ± 0.01 ^a^	0.44 ± 0.08 ^b^	0.16 ± 0.04 ^bc^	0.2 ± 0.05 ^bc^	60	<0.001 ***	0.078
*P. agardhii*	build up	0.12 ± 0.01 ^a^	0.16 ± 0.01 ^a^	0.51 ± 0.06 ^bc^	0.39 ± 0.06 ^bd^	72	<0.001 ***	<0.001 ***
	decline	0.04 ± 0.004 ^a^	0.25 ± 0.08 ^b^	0.19 ± 0.07 ^a^	0.08 ± 0.02 ^a^	60	0.017 *	0.076

^1^ Two-way repeated measures analysis of variance (RM ANOVA) (grouping factor). *** *p* < 0.001, ** *p* < 0.01, * *p* < 0.05; ^2^ Superscript letters (^a,b,c,d^) indicate subgroups of non-AA treatments that were not significantly different based on post hoc pairwise comparison (Bonferroni test, *p* > 0.05); ^3^ Two-way RM ANOVA (time factor).

**Figure 7 toxins-17-00278-f007:**
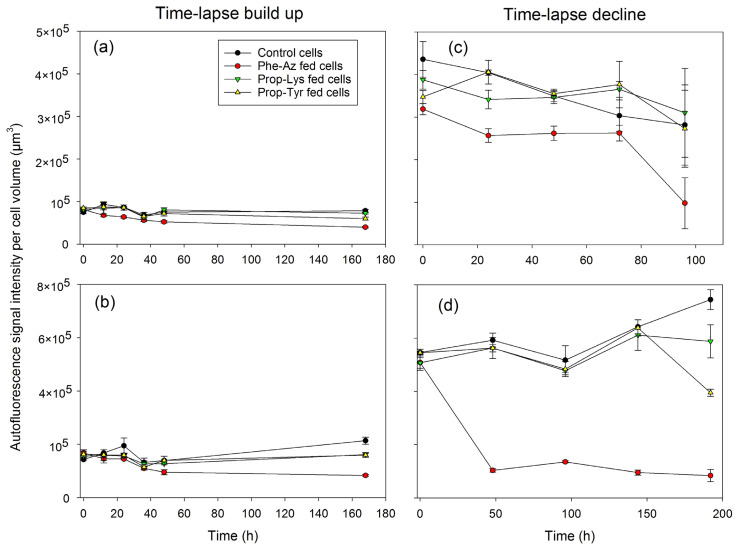
The mean (±SE) autofluorescence intensity per cell volume (in µm^3^) of *M. aeruginosa* (**a**,**c**) or *P. agardhii* (**b**,**d**) during time-lapse experiments related to the build up (**a**,**b**) or decline (**c**,**d**) of clickable MC (**a**,**c**) or clickable AP (**b**,**d**). AF signal intensities were quantified from images, as shown in Figure 2, Figure 3, Figure 4 and Figure 5.

### 2.4. Relationship Between Clickable Cyanotoxin Content and Labeling by the Click Reaction as Quantified from High-Resolution Microscopy

Because aliquots of cells were analyzed for their AP/MC content, as described in a previous study [23], the A488 signal intensity results were compared directly. Based on the original data, there were significant linear relationships between the A488 signal intensity and the percentage of clickable MC/AP based on the total MC/AP content or µg of clickable MC/AP per mg of dry weight (DW). Overall, the log(x + 1)-transformed data showed higher coefficients of determination (R^2^), i.e., the proportion of variation in the A488 signal intensity explained by the clickable MC/AP content in the linear regression model. All reported regression curves were statistically significant (*p* < 0.001 for both intercept and slope, sample size [*n*] = 72 time-lapse build up, *n* = 60 for time-lapse decline; Figure 8 and Appendix A).

In general, the A488 labeling intensity data for the three non-AA treatments consistently corelated to the observed clickable MC/AP content. The control cells—grown and processed under identical conditions but without non-AA substrate—always showed minimum A488 intensity. This minimum intensity probably resulted from background fluorescence given that a previous study provided no evidence for non-specific binding of the A488 fluorophore for the same *M. aeruginosa* strain [22]. For *M. aeruginosa*, Prop-Tyr feeding led to the highest clickable MC-Prop-Tyr content along with the highest A488 signal intensity (Figure 8a,c and Appendix A). On the contrary, the Prop-Lys-fed cells showed lower A488 labeling intensity, consistent with the lowest overall proportion of clickable MC-Prop-Lys in total MC. The Phe-Az-fed cells showed a more variable and intermediate A488 labeling intensity and proportion of MC-Phe-Az in total MC.

For *P. agardhii*, the highest A488 labeling intensity correlated to the highest percentage of AP-Prop-Lys in the peptide extract (Figure 8b,d and Appendix A). For Prop-Tyr-fed cells, there was a lower proportion of clickable AP-Prop-Tyr that corresponded to a minimum or lower A488 signal intensity. In contrast, synthesis of clickable AP-Phe-Az was observed, but it did not result in any A488 labeling intensity. Because the growth of Phe-Az-fed cells was reduced dramatically, the clickable AP-Phe-Az content in µg per DW also was also low (Appendix A). Thus, toxic side effects caused by Phe-Az feeding in *P. agardhii* not only reduced cell division and growth [23], but also dramatically reduced AF and A488 labeling intensity.

**Table 3 toxins-17-00278-t003:** The mean (±SE) of co-localization coefficients between A488 signal intensity vs. autofluorescence (AF) signal intensity of *M. aeruginosa* or *P. agardhii* during time-lapse experiments using pulsed feeding of non-AAs to observe the build up or decline of clickable MC or AP. Graphical data are shown in Appendix A. The measurements for each time point are shown in Appendix A.

Strain	Experiment	Control	Phe-Az	Prop-Lys	Prop-Tyr	*n*	*p*-Value ^1,2^ (Treatment)	*p*-Value ^3^ (Time)
Object Pearson’s co-localization coefficient								
*M. aeruginosa*	build up	0.22 ± 0.01	0.18 ± 0.01	0.21 ± 0.02	0.2 ± 0.01	72	0.288	0.03 **
	decline	0.36 ± 0.01	0.32 ± 0.01	0.35 ± 0.01	0.34 ± 0.01	60	0.105	0.006 **
*P. agardhii*	build up	0.23 ± 0.02 ^a^	0.2 ± 0.02 ^ab^	0.19 ± 0.02 ^ab^	0.17 ± 0.02 ^b^	72	0.022 *	0.003 **
	decline	0.47 ± 0.03 ^a^	0.14 ± 0.09 ^b^	0.25 ± 0.03 ^ab^	0.18 ± 0.03 ^b^	60	0.006 **	0.229
Object Spearman’s co-localization coefficient								
*M. aeruginosa*	build up	0.41 ± 0.02	0.37 ± 0.02	0.39 ± 0.03	0.39 ± 0.02	72	0.727	0.006 **
	decline	0.64 ± 0.01 ^a^	0.61 ± 0.01 ^b^	0.63 ± 0.01 ^ab^	0.63 ± 0.01 ^ab^	60	0.035 *	0.532
*P. agardhii*	build up	0.68 ± 0.02 ^a^	0.67 ± 0.02 ^a^	0.53 ± 0.03 ^b^	0.46 ± 0.04 ^b^	72	0.001 ***	<0.001 ***
	decline	0.54 ± 0.04 ^a^	0.22 ± 0.11 ^b^	0.56 ± 0.03 ^a^	0.44 ± 0.05 ^ab^	60	0.013 *	0.258

^1^ Two-way repeated measures analysis of variance (RM ANOVA) (grouping factor). *** *p* < 0.001, ** *p* < 0.01, * *p* < 0.05; ^2^ Superscript letters (^a,b^) indicate subgroups of non-AA treatments that were not significantly different based on post hoc pairwise comparison (Bonferroni test, *p* > 0.05); ^3^ Two-way RM ANOVA (time factor).

**Figure 8 toxins-17-00278-f008:**
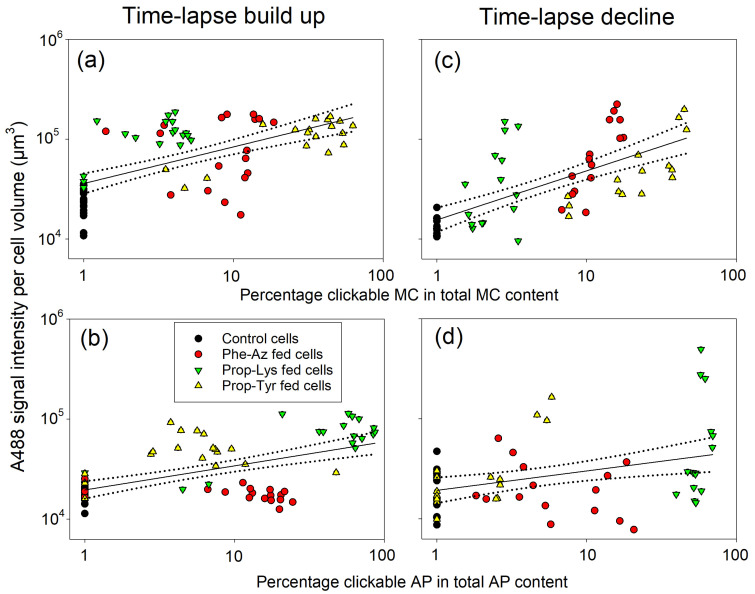
Alexa Fluor 488 (A488) signal intensity per cell volume (in µm^3^) of *M. aeruginosa* (**a**,**c**) or *P. agardhii* (**b**,**d**) during time-lapse experiments using pulsed feeding of non-AAs vs. the percentage of clickable MC (**a**,**c**) or clickable AP (**b**,**d**), as recorded by chemical-analytical methods reported previously [23]. Details of linear regression lines: (**a**) Y = a + bx, where y is log (x + 1) A488 signal intensity and x is log(x + 1) percentage of clickable MC/AP in total MC/AP as determined by liquid chromatography–mass spectrometry [23]. The dotted lines indicate 95% confidence intervals. Regression curves: (**a**) y = 4.524 + 0.374x (R^2^ = 0.46, *p* < 0.001), (**c**) y = 4.193 + 0.493x (R^2^ = 0.48, *p* < 0.001), (**b**) y = 4.289 + 0.243x (R^2^ = 0.33, *p* < 0.001), (**d**) y = 4.281 + 0.198x (R^2^ = 0.14, *p* = 0.004), without Phe-Az-fed cells (**b**) y = 4.321 + 0.329x (R^2^ = 0.68, *p* < 0.001), (**d**) y = 4.308 + 0.233x (R^2^ = 0.22, *p* = 0.001). MC/AP contents for each time point are shown in Appendix A.

### 2.5. Relationship Between the Clickable Cyanotoxin Content and Labeling by the Click Reaction as Quantified by FCM

In addition to high-resolution microscopy, FCM was used to estimate the number of A488-labeled cells/filaments resulting from chemo-selective labeling. Because FCM easily enumerates thousands of particles (cells) to discriminate labeled from non-labeled subpopulations, it was applied as a complementary and more quantitative approach compared with microscopic observation. During the time-lapse build up experiment, labeled subpopulations appeared at T0, showing an increase in BL1-A fluorescence (Appendix A). During the time-lapse signal build up, labeled subpopulations of Prop-Lys- and Prop-Tyr-fed cells differentiated gradually from control cells. In other words, the overlap in the frequency distribution curves for individual subpopulations decreased over time. From T1 onwards, the maximum particle frequency of the BL1-A (A488) signal varied by 1–2 orders of magnitude from the maximum BL1-A frequency for the control cells. In contrast, the unlabeled control cells always showed a narrow range of BL1-A fluorescence intensity. During the time-lapse decline experiment, the BL1-A fluorescent subpopulation also declined and became undetectable compared with the rather sharp fraction of unlabeled control cells (Appendix A).

Corresponding to A488 intensity per cell volume, the BL1-A percentage of cells correlated positively to the proportion of clickable MC/AP in the total MC/AP content (Figure 9) or µg of clickable MC/AP content per mg of DW (Appendix A). For *M. aeruginosa*, Prop-Tyr-fed cells presented the highest clickable MC content, which correlated with a high proportion of BL1-A particles. Similarly, Phe-Az-fed cells showed a positive relationship between clickable MC synthesis and the percentage of BL-1A fluorescent particles (Figure 9a,c). In contrast, for *P. agardhii*, Phe-Az fed cells incorporated Phe-Az into modified AP but did not show increased fluorescence compared with controls (Figure 9b,d). Given that AF was also dramatically reduced in Phe-Az-fed cells (Figure 7), it is concluded that Phe-Az had the strongest toxic side effects in *P. agardhii*.

**Figure 9 toxins-17-00278-f009:**
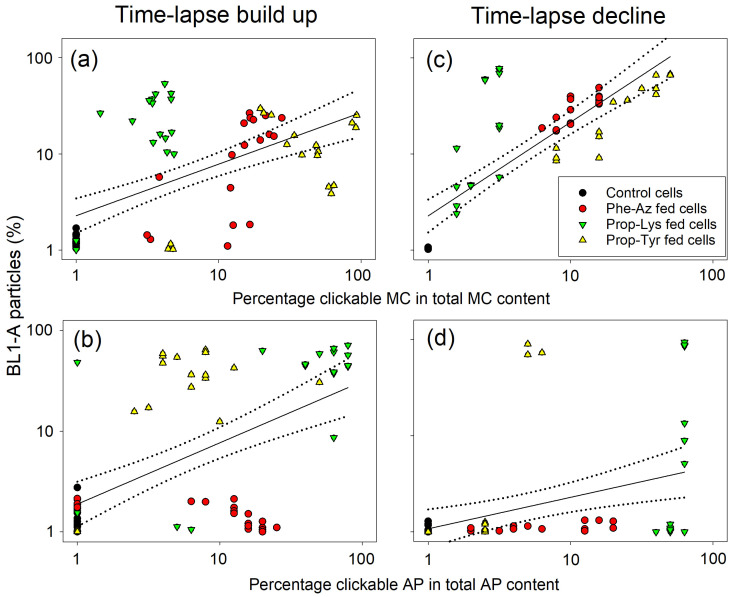
BL1-A percentage of cells (A488-labeled particles) of *M. aeruginosa* (**a**,**c**) or *P. agardhii* (**b**,**d**) during time-lapse experiments vs. the percentage of clickable MC (**a**,**c**) or AP (**b**,**d**) as recorded by flow cytometry. Details of linear regression lines: (**a**) Y = a + bx, where y is log (x + 1) A488 particles and x is log(x + 1) percentage of clickable MC/AP in total MC/AP as determined by liquid chromatography–mass spectrometry [23]. The dotted lines indicate 95% confidence intervals. Regression curves: (**a**) y = 0.36 + 0.54x (R^2^ = 0.33, *p* < 0.001), (**c**) y = 0.36 + 0.97x (R^2^ = 0.64, *p* < 0.001), (**b**) y = 0.23 + 0.64x (R^2^ = 0.35, *p* < 0.001), (d) y = 0.03 + 0.33x (R^2^ = 0.15, *p* = 0.002), without Phe-Az-fed cells (**b**) y = 0.365 + 0.825x (R^2^ = 0.62, *p* < 0.001), (**d**) y = 0.091 + 0.36x (R^2^ = 0.18, *p* = 0.004). MC/AP contents for each time point are shown in Appendix A.

## 3. Discussion

### 3.1. Sensitivity of Chemo-Selective Labeling and Visualization via A488

In general, cyanobacterial AF can disturb fluorescence signal detection because of spectral overlap with commonly used fluorophores, including Cy3, Cy5, and FITC [29]. While methods to improve the signal-to-noise ratio have been developed continuously [30,31], the choice of the appropriate fluorophore (i.e., A488) is of the utmost importance [32]. In the present study, the A488 labeling signal responded sensitively, which is consistent with the early synthesis of clickable MC/AP as observed in peptide extracts reported earlier [23]. During the time-lapse increase experiment, the A488 signal could be reliably differentiated from background fluorescence (i.e., AF) as early as T1 (after 12 h). For both *M. aeruginosa* and *P. agardhii*, the AF signal intensity (i.e., emission spectrum 680–730 nm) was generally higher than the signal for A488 (i.e., 500–550 nm), (Table 2 and Appendix A).

Both *M. aeruginosa* and *P. agardhii* contained low amounts of clickable MC/AP as early as T0: *M. aeruginosa*, 5% ± 1% for Phe-Az and 4% ± 1% for Prop-Tyr; *P. agardhii*, 3% ± 2% for Prop-Lys (see Tables S4 and S5, and Figures 1 and 2 in [23]). The early incorporation of non-AA at T0 can be explained by the manipulation time (approximately 1 h) including centrifugation and washing in phosphate-buffered saline (PBS) until fixation in 2% paraformaldehyde (PFA). These low percentages, however, did not result in a significant increase in A488 signal intensity between treatments based on two-way repeated measures analysis of variance (RM ANOVA) and post hoc pairwise multiple comparisons (Bonferroni *t*-test, *p* > 0.95, Figure 6). Consequently, to determine the detectability vs. the A488 signal-to-noise ratio more precisely, another time-lapse build up experiment in hourly time intervals would be needed. There was no difference in the timing of clickable MC vs. AP synthesis between the two genera *M. aeruginosa* and *P. agardhii*, which can be explained at least partly by the comparable growth rates of these strains, that is, r = 0.35–0.6 and 0.2–0.4 day^−1^, respectively (see Tables 1 and 2 in [23]).

FCM was more sensitive than high-resolution microscopy, as already at T0 during the time-lapse build up experiment labeled subpopulations were detected in the Prop-Tyr and Prop-Lys fed cells (Appendix A). In FCM, the rectangular gating tool allows one to discriminate relatively low numbers of particles compared with control cells, supporting the early discrimination of labeled from non-labeled subpopulations. Under the specified conditions, during time-lapse build up, only 27–149 “false” positive particles for *M. aeruginosa* and 0–33 “false” positive particles for *P. agardhii* were differentiated, resulting in 0.04–0.69% and 0–1.75% of the total population, respectively (*n* = 18). For FCM, the observed noise implies a limit of detection of around 1%. In contrast, the natural A488 signal intensity per cell volume recorded from the control cells in the microscope showed greater variation: During time-lapse build up, it was 50–150% for *M. aeruginosa* and 75–118% for *P. agardhii* (*n* = 18). Thus, the potential limit of detection for A488 labeling is a 1.5-fold and 1.2-fold excess of the natural A488 intensity in *M. aeruginosa* and *P. agardhii*, respectively.

So far, only cells treated chemically (i.e., fixed in 2% PFA and permeabilized using Triton X-100, Bio-Rad, Vienna, Austria) have been used [21]. While this protocol has the advantage of processing aliquots for different instrumentation simultaneously (e.g., cell numbers, chemical analysis, microscopy, and FCM), in the future live cell observation would be advisable in combination with FCM. In this study, the BL1-A fluorescent *M. aeruginosa* cells with clickable MC-Prop-Tyr fluctuated from T1 to T5, implying that clickable MC-Prop-Tyr was not produced constantly in the cells from one time point to the next (Figure 9, Appendix A). In addition, the distribution curve form of histograms showing the relative frequency (%) of BL1-A fluorescent particles changed during the time-lapse increase (Appendix A). Catalyst-free cycloadditions that avoid toxic copper side effects, such as the strain-promoted azide–alkyne cycloaddition (SPAAC) [33,34], have been developed but applied less frequently. Such protocols could make it possible to monitor the same cells of an experimental population repeatedly.

### 3.2. Quantitative Relationship Between the Clickable Cyanotoxin Content and A488 Labeling Signal by the Click Reaction

Overall, the early visibility of clickable MC/AP is in accordance with other studies on the NRPS synthesis rate. Moschny et al. [35] reported that, when using precursor-directed biosynthesis of MC in *M. aeruginosa*, the amount of clickable MC extracted from the biomass most likely increased in an exponential manner during the first 36 h, reaching a maximum at 48 h. Accordingly, for heterologously expressed NRPS domains, researchers have observed significant amounts of clickable peptide products within a few hours [36]. In penicillin biosynthesis, for the NRPS enzyme (ACV synthetase) forming the tripeptide ACV δ-(L-R-aminoadipyl)-L-cysteinyl-D-valine, under optimal conditions in vitro, a linear rate of ACV formation was observed for at least 120 min [37]. Together with our previous report [23], a chemical rationale has been provided by (i) analyzing aliquots via high-performance liquid chromatography–mass spectrometry and protonated mass (MS^2^/MS^3^) fragmentation analysis for the clickable MC/AP content and (ii) the subsequently created A488 signal in the cells using the click reaction (Appendix A).

For *M. aeruginosa*, the proportion of clickable MC obtained through all three non-AAs correlated to the A488 signal intensity per cell (Figure 8a,c) and BL1-A fluorescent particles as recorded by FCM (Figure 9a,c). When testing Phe-Az or Prop-Lys individually, there were significant linear relationships between the percentage of clickable MC vs. the A488 signal intensity (T0–T5, *n* = 18). In contrast, for Prop-Tyr, the highest A488 signal intensity was recorded from T1 onwards, consistent with the fact that clickable MC-Prop-TyrR showed the most efficient synthesis. The variable but consistent differences in clickable MC/AP synthesis can be explained by the variable catalytic promiscuity among different NRPS genotypes [23].

For *P. agardhii*, the A488 signal intensity per cell volume also correlated to the proportion of clickable AP-Prop-Lys and AP-Prop-Tyr in total AP (Figure 8). For the entire population, the proportion of the labeled subpopulation also correlated with the clickable AP-Prop-Lys or AP-Prop-Tyr proportion (Figure 9). In contrast, there was no such relationship for AP-Phe-Az. Recently, toxic side effects have been reported for *Pseudomonas aeruginosa* when using 4-azido-l-homoalanine (1.5 mM concentration in the medium) to modify the siderophore pyoverdine via NRPS [38]. The possible reasons for azide-based toxicity have been discussed [23], including the well-known, exceptional reactivity of the azide group [39]. Despite evidence of incorporation into MC/AP, Phe-Az also decreased AF significantly in both cyanobacteria (Figure 7). Thus, one possible reason for reduced AF is intracellular decomposition of Phe-Az resulting in reactive nitrenes causing the overall toxic side effects. Nevertheless, for *P. agardhii*, even under toxic conditions, the modified peptide AP-Phe-Az was still synthesized at an intermediate proportion (12.5 ± 1.7%, ranging from 5.6–20.5%) relative to AP-Prop-Lys (50 ± 6.8%, ranging from 3.5–85.4%) and AP-Prop-Tyr (6.4 ± 2.5%, ranging from 1.8–10.8%) (Tables S4 and S5 and Table 3 in [23]). In summary, although Phe-Az is incorporated into MC/AP to some extent, most of the non-incorporated Phe-Az in the cell might have resulted in the formation of nitrenes via uncontrolled decomposition processes counteracting the expected A488 fluorescence signal. Obviously, *M. aeruginosa* was less sensitive to Phe-Az than *P. agardhii* despite showing high intracellular Phe-Az content (Figure S9 in [23]), implying genus-specific differences in detoxification mechanisms.

### 3.3. Spatiotemporal Changes in the A488 Signal in Cyanobacteria During the Time-Lapse Decline Experiment

As reported previously, there was a rather heterogeneous distribution of the A488 signal for clickable MC/AP products; this signal was quantified in relation to AF using standard co-localization indices, namely Object Pearson’s and Object Spearman’s co-localization coefficients (Table 3 and Appendix A). For *M. aeruginosa*, the co-localization of A488 signal vs. AF on average tended to decrease (although the decrease was not significant), indicating the presence of distinct A488 signal entities in the cell. In contrast, for *P. agardhii* both co-localization indices of A488 signal vs. AF decreased significantly as a result of A488 signal build up and decline. This heterogeneity could be explained by (i) structures facilitating MC/AP accumulation in the cell and/or (ii) MC/AP biosynthesis. For the former, it has been argued that the intracellular MC/AP content sometimes exceeds the theoretical solubility threshold [40]. A dual labeling protocol has been used to test whether lipids might be related to MC solubility and thus spatially related to clickable MC accumulation occurring as distinct entities [22]. However, pairwise co-localization indices between both lipids (visualized using Bodipy staining) and clickable MC (A405) were not increased compared with analogous indices for AF. The two hypotheses, accumulation vs. biosynthesis of MC/AP in *M. aeruginosa* and *P. agardhii*, will be tested in another forthcoming study via advanced image analysis.

Interestingly, in Prop-Lys- and Prop-Tyr-fed *P. agardhii*, the distinct entities located towards the cell septa were replaced gradually by ring-like formations in the peripheral area of the cell (Figure 5). It is tempting to speculate that the ring-like formations originated from distinct entities (i.e., representing AP-Prop-Lys or AP-Prop-Tyr) as leftovers from diffusion because the synthesis of AP-Prop-Lys/Tyr stopped 48 h earlier. In contrast, during time-lapse decline, there was no such fading signal for *M. aeruginosa*. Given that *M. aeruginosa* showed distinct entities distributed all over the cell, it might be more difficult to see the diffusing clickable MC because of a lower spatial concentration. Notably, using a fluorescence in situ hybridization probe specific for the mRNA transcribed from the *mcy*A gene part of the *mcy* gene cluster in *M. aeruginosa* PCC7806, a so-called “nonuniform labelling, concentrated in spots” has been reported [41,42]. The authors concluded that their fluorescence technique allowed the visualization of the variation in the expression of the *mcy* synthesis genes in *M. aeruginosa*. Using the same fluorescent probe, non-uniform labeling results have been reproduced using *M. aeruginosa* strain PCC7806 [43]. It has been suggested repeatedly that MC carrying *N*-methyl-dehydroalanine (Mdha) at position 7 of the MC molecule binds covalently to abundant proteins, possibly protecting them from photooxidation, and undergoes several other molecular interactions in the *M. aeruginosa* cell [44]. Initial attempts have been made to test co-localization of clickable MC-Prop-Tyr and abundant proteins (e.g., using RbcL and FtsZ [cytosolic proteins] and PsbA [a thylakoid protein]) in *M. aeruginosa*. A double labeling protocol was applied by immunolabeling of proteins (green A488) with chemo-selective labeling of clickable MC (blue A405) and revealed increased co-localization coefficients between PsbA and AF, but not between clickable MC and immunolabeled proteins [22]. In the future, predictions about various cell structures/organelles interacting with cyanotoxins or not could be tested via chemo-selective labeling.

### 3.4. The Potential of Visualizing Cyanotoxin Synthesis via Chemo-Selective Labeling in Gene or Peptide Function Studies

The application of fluorescence-based technology for MC detection in the environment, cells, or other biota has been intensively tested [43,45]. In general, the feeding of clickable non-AAs or precursor-directed synthesis of clickable MC/AP in the environment is considered to be less applicable, and one might think, rather, of functional research applications—for example, via heterologous expression systems [46]. For example, heterologous expression of MC-LR in *Escherichia coli* was first reported in 2017 [47], and then more recently in a model cyanobacterium *Synechococcus elongatus* PCC7942 [48]. For *E. coli*, the large MC biosynthesis (NRPS) gene cluster (55,000 base pairs) was cloned using fosmids and heterologously expressed in *E. coli*. Notably, the natural bidirectional promotor was not sufficient for any detectable MC product. Only the introduction of a strong bidirectional tetracycline-inducible promotor led to the ability to detect the corresponding MC product [49]. Through chemo-selective labeling and cyanotoxin synthesis on the cellular level in the host organism, such as *E. coli,* the MC production process could be observed directly. Such a heterologous expression system has also been used to investigate the biosynthetic flexibility of the MC biosynthesis pathway [50] or to test the functions of accessory genes [51]. A flexible MC biosynthesis pathway can help to produce clickable MC structural variants with specific moieties to compare moiety-dependent biological properties, such as eukaryotic cellular uptake rates [52]. Another application might involve an investigation of the role of environmental factors, such as macronutrients and light conditions under standardized heterologous expression conditions. For example, soon after the elucidation of the MC biosynthesis pathway [5], binding boxes of the ubiquitous transcription factors NtcA (a transcriptional regulator of genes involved in nitrogen assimilation) and Fur (a ferric uptake regulator) were identified within the bidirectional promotor region and could be tested using chemo-selective labeling more directly under standardized in vivo conditions. In addition, the direct effects of irradiance (i.e., high vs. low light) regulated the transcription of the MC biosynthesis genes in *Microcystis* [44,53]. In summary, chemo-selective labeling should allow for the ability to monitor the molecular basis of potential factors in regulation more directly, in real time, and with high temporal resolution.

## 4. Conclusions

In conclusion, time-resolved analysis of the A488 signal intensity points to a rather specific and sensitive CuAAC reaction between clickable modified toxins/peptides and the A488 signal based on qualitative and quantitative correlation. On a spatial scale during the time-lapse increase experiment, the heterogeneous A488 signal structures, described as distinct entities previously, became visible from the beginning and varied in their location on a subcellular level. On a temporal scale during the time-lapse decline experiment, the heterogeneous structures in the cell either faded or were gradually replaced by a ring-like patch formation in the peripheral area of the cell. Thus, real-time observation of cyanotoxin synthesis by means of click chemistry has the potential to track intracellular synthesis using both chemical–analytical analysis and high-resolution imaging and can be used to study regulation and potential intracellular function.

## 5. Materials and Methods

### 5.1. Study Organisms and Experimental Design

As described previously, clonal strain cultures of *M. aeruginosa* strain Hofbauer and *P. agardhii* strain no371/1 were grown semi-continuously under maximum growth conditions in BG11 medium [54] at 20 °C using illumination conditions of 50 µmol photons m^−2^ s^−1^ (16:8 h light/dark cycle) [23]. Details of strain origin and identification were reported [21].

For time-lapse signal build up, at T0 the treatments were supplemented once with Phe-Az or Prop-Lys or Prop-Tyr, resulting in a final concentration of 0.05 mM [23] (Figure 10). Control cell cultures were grown under identical conditions but in the absence of non-AAs. At T0, the cells were harvested directly after adding the non-AA to the medium within 1 h. To observe the change in the A488 signal intensity, culture flasks were harvested every 12 h for the first 48 h (T1–T4), with a final harvesting at T5 (168 h).

For time-lapse signal decline, cultures were grown for 48 h in the presence of one of the three non-AAs. At T0, following 48 h of non-AA feeding incubation, the cultures were washed by centrifugation (4000 rpm for 5 min) and transferred to fresh (non-AA free) BG11 medium. Analogously to the build up experiment, at T0, the cells were harvested after transfer into new medium within 1 h. The time points T1–T4 represented every 24 h for *M. aeruginosa* and every 48 h for *P. agardhii*. All harvest steps were performed at room temperature.

**Figure 10 toxins-17-00278-f010:**
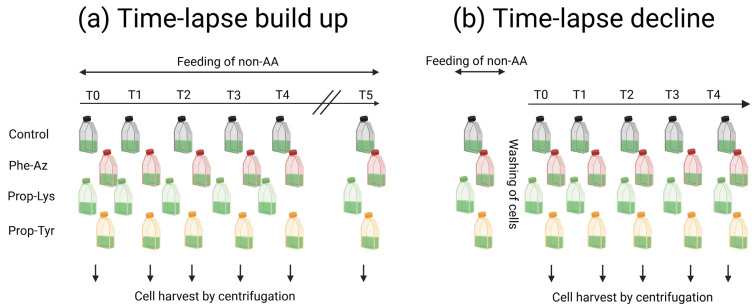
Experimental design for time-lapse experiments to allow for real-time observation of clickable MC/AP synthesis in bloom-forming cyanobacteria: (**a**) time-lapse build up experiment and (**b**) time-lapse decline experiment. Three technical replicates were measured for each time point and non-AA treatment. Note that the corresponding clickable MC/AP contents as recorded from aliquots, as well as the growth rates, have been reported previously [23]. Created in BioRender. Deng, L. (2025) https://BioRender.com/yvkfgrd.

### 5.2. Cell Harvesting and Fixation

Cells from 40-mL culture flasks were harvested by centrifugation (4000× *g*, 15 min) after breaking the gas vesicles using the hammer, cork, and bottle method. The detailed protocol has been published elsewhere [21]. In brief, the cells were washed with freshly prepared 1× PBS, fixed with 2% PFA for 15 min, washed, incubated in Triton X-100 (0.1% in PBS, *v*/*v*) for 10 min, washed with PBS, and stored in 50 µL PBS at 4 °C for the click reaction (see Section 5.3) the following day.

In parallel, cells were filtered on glass fiber filters by vacuum filtration and extracted for natural and clickable peptides as described previously [23]. The quantitative data on clickable cyanopeptide synthesis as recorded via liquid chromatography–mass spectrometry from aliquots obtained during the same experiments have been reported [23]. In this study the same data were used to explore the relationship between the clickable peptide content and labeling by the click reaction.

### 5.3. Chemo-Selective Labeling of MC/AP Synthesis via the Click Reaction

The cells were fixed the day before and stored in 1× PBS at 4 °C in the dark until use for the CuAAC reaction using a commercial chemical reaction buffer (C10269, Fisher Scientific, Vienna, Austria) containing the reaction buffer, a buffer additive, 100 mM copper sulfate solution, Milli Q water, and the fluorophore, namely A488 azide (A10266, Fisher Scientific) for non-AA treatments Prop-Lys and Prop-Tyr or A488 alkyne (A10267, Fisher Scientific). A488 azide/alkyne is a photostable and bright green fluorescent dye with excitation/emission maxima of 495/519 nm and with a molecular weight of 861.04/773.91 Da. The reaction volumes and components of the click chemical reaction were as follows: 200 µL of MQ water, 22 µL of (10×) reaction buffer (containing Tris-buffered saline), 5 µL of 100 mM CuSO_4_ (reduced state), 25 µL of (10×) buffer additive, and 1 µL of A488 azide/alkyne (1 mM aliquots stored at −20 °C).

Cyanobacterial pellets were incubated in PBS containing 2% bovine serum albumin (BSA) for 30 min at room temperature, and subsequently transferred into the CuAAC chemical reaction buffer for 60 min in the dark at room temperature, vortexing every 10 min. Then, the samples were centrifuged at 8000 rpm for 10 min, and the supernatant was discarded. Finally, the cells were washed twice, the first time with PBS containing 2% BSA washing solution and the second time with PBS only. The cells were dispersed on coverslips (24 mm × 24 mm, 0.17 ± 0.01 mm, Hecht Assistant, Sondheim vor der Rhön, Germany) and briefly air-dried before they were embedded in ProLong Diamond Antifade Mountant (Fisher Scientific) for 48 h in the dark.

### 5.4. Microscopic Recording of Signal Labeling

Imaging was performed as described earlier [21,22] with a high-resolution laser scanning microscope DMI 6000 SP8 (Leica Microsystems, Wetzlar, Germany) with a 100× HCX Plan Apo 1.4 oil objective. The emission spectral range in channel 0 (A488) was narrowed down from 504 to 580 nm to 500–550 nm to enable the visualization of another fluorophore (i.e., to allow for dual labeling). Images were acquired at an XY resolution of 50 nm and Z resolution of 150 nm (resulting in 30–40 scans in depth per cell). In general, during image acquisition care was taken to prevent oversaturation conditions: the power of the white light laser (WLL) was adjusted prior to scanning the cell. Additionally, the Leica motion correction collar was adjusted independently for each microscopic slide to increase image quality and to reduce refractive index mismatch caused by the cover slip. Cells for imaging were selected randomly. For each time point and treatment, 20 randomly selected cells were viewed for each of three technical replicates.

All images were deconvolved with Huygens Essential version 23.10 (Scientific Volume Imaging, Hilversum, The Netherlands, http://svi.nl), using the default conditional maximum likelihood estimation algorithm. The total intensity per cell was compared for two channels (channel 0, A 488, 498 nm excitation; channel 1, AF, 620 nm).

### 5.5. Calculation of Co-Localization Indices

Two co-localization indices were used as a statistical measure of the spatial relationship between the two channels: Pearson’s linear correlation coefficient and Spearman’s coefficient based on the intensity ranks instead of intensity values. Both coefficients indicate how strongly two variables correlate with each other: A value of +1 indicates a perfect positive correlation, while a value of −1 means a perfect negative correlation. To minimize the background influence, Object Pearson’s co-localization coefficient and Object Spearman’s co-localization coefficient were used [55,56]. All co-localization parameters were estimated in Huygens Essential version 23.10.

### 5.6. FCM Analysis

Standard FCM was used to quantify the number of stained cells/filaments resulting from the chemo-selective click reaction. The particles were recorded using the Attune Acoustic Focusing Cytometer (Life Technologies, Thermo Fisher Scientific, Darmstadt, Germany) via the blue (BL, 488 nm, 50 mW) solid state laser. Forward scatter and BL1-A fluorescence (band pass filter at 530 ± 15 nm) were used for all measurements to discriminate stained and non-stained populations of cells at a flow rate of 0.1 mL min^−1^. The acquisition volume was 0.5 mL. The target populations were quantified by rectangular gating on forward scatter vs. BL-1A fluorescence as compared with control cells.

### 5.7. Statistical Analysis

Three technical replicates were measured for each time point and non-AA treatment. The cell diameter and cell volume (Table 1) and A488 signal intensity, AF, and the A488/AF ratio (Table 1, Table 2, Table 3 and Appendix A) were compared with two-way RM ANOVA (consisting of the grouping factor: three non-AA treatments and one control; and the time factor: T0–T4 (T5), three replicates). The time factor was always significant (*p* < 0.001). The assumption of normality (based on the Shapiro–Wilk) was not met in part for the A488 intensity data and not met for the A488/AF ratio data, while the equality of variance (based on the Brown–Forsythe test) was always met (*p* < 0.05). Because ANOVA results are generally considered robust if the sample sizes between treatments are equal, it was still used for the analysis. The Bonferroni *t*-test was used for the post hoc pairwise multiple comparison procedure, with *p* < 0.05 considered to indicate a significant difference.

For linear regression analysis, the data on clickable MC/AP in the total MC/AP content and the A488 signal intensity as recorded by both microscopy and FCM were log(x + 1) transformed to make the data linear and to meet the assumptions of normality (based on the Shapiro–Wilk test) and a constant variance. The linear regression curves between A488 signal intensity (or % Bl1-A particles as recorded by FCM) and the percentage of clickable MC/AP in the total MC/AP content (Figure 8 and Figure 9) or µg of clickable MC/AP per mg of DW (Appendix A) were calculated. The sample size was 72 for time-lapse signal build up experiments and 60 for time-lapse signal decline experiments. For data that did not meet the assumptions of a normal distribution or equal variance, the Spearman rank-order correlation was calculated, with a significant positive correlation in each case. All tests were performed using Sigma Plot for Windows version 14.0.

### 5.8. Chemical Analytical Analysis

As reported previously [21,22,23], the original MC/AP structural variants and the new (clickable) MC/AP structural variants were differentiated and quantified from aliquots and are summarized in Appendix A. In *M. aeruginosa* strain Hofbauer, the original MC was composed of demethylated MC variants DAsp-MC-YR [M + H]^+^ 1031.5 and D-Asp-MC-LR [M + H]^+^ 981.5 and methylated MC variants MC-YR [M + H]^+^ 1045.5 and MC-LR [M + H]^+^ 995.5, which were partly transformed into clickable MC. For Phe-Az, the clickable variants DAsp-MC-Phe-AzR [M + H]^+^ 1056.5 and MC-Phe-AzR [M + H]^+^ 1070.5, for Prop-Lys the variants DAsp-MC-Prop-LysR [M + H]^+^ 1078.5 and MC-Prop-LysR [M + H]^+^ 1092.5, and for Prop-Tyr the variants DAsp-MC-Prop-TyrR [M + H]^+^ 1069.5 and MC-Prop-TyrR [M + H]^+^ 1083.5 were reported [21,22,23]. In *P. agardhii* strain no371/1, the original AP structural variants included AP C [M + H]^+^ 809.5, AP B [M + H]^+^ 837.5, and AP A [M + H]^+^ 844.5. The clickable APs included AP-Phe-Az [M + H]^+^ 843.3, AP-Prop-Lys [M + H]^+^ 891.5, or AP-Prop-Tyr [M + H]^+^ 882.5.

## Figures and Tables

**Table 1 toxins-17-00278-t001:** The mean (±standard error (SE) of the mean) cell diameter and cell volume during time-lapse signal build up and decline in *M. aeruginosa* strain Hofbauer and *P. agardhii* strain no371/1. Control = cells grown and processed under identical conditions but without non-AA substrate. Graphical data are shown in Appendix A. Measurements for each time point are shown in Appendix A.

Strain	Experiment	Control	Phe-Az	Prop-Lys	Prop-Tyr	*n*	*p*-Value ^1,2^ (Treatment)	*p*-Value ^3^ (Time)
Cell diameter (µm)								
*M. aeruginosa*	build up	4.3 ± 0.04 ^a^	4.59 ± 0.08 ^b^	4.42 ± 0.04 ^c^	4.47 ± 0.04 ^c^	72	0.001 ***	<0.001 ***
	decline	4.05 ± 0.03 ^a^	4.59 ± 0.1 ^b^	4.23 ± 0.03 ^c^	4.14 ± 0.03 ^ac^	60	<0.001 ***	0.008 **
*P. agardhii*	build up	3.52 ± 0.02	3.56 ± 0.02	3.53 ± 0.02	3.52 ± 0.02	72	0.084	<0.001 ***
	decline	3.58 ± 0.05 ^a^	3.74 ± 0.03 ^b^	3.58 ± 0.04 ^a^	3.49 ± 0.06 ^a^	60	0.001 **	0.197
Cell volume (µm^3^)								
*M. aeruginosa*	build up	42.4 ± 1.1 ^a^	52.4 ± 2.9 ^b^	46.2 ± 1.3 ^c^	47.8 ± 1.4 ^c^	72	<0.001 ***	<0.001 ***
	decline	35.3 ± 0.8 ^a^	53.1 ± 3.9 ^b^	40.3 ± 0.9 ^c^	38 ± 0.7 ^ac^	60	<0.001 ***	0.008 **
*P. agardhii*	build up	133.3 ± 1.6	137.2 ± 2	134 ± 1.3	134 ± 1.4	72	0.083	0.01 *
	decline	140.5 ± 3.8 ^a^	154 ± 2.8 ^b^	140.5 ± 3.5 ^a^	133 ± 4.2 ^a^	60	0.002 **	0.398

^1^ Two-way repeated measures analysis of variance (RM ANOVA) (grouping factor). *** *p* < 0.001, ** *p* < 0.01, * *p* < 0.05; ^2^ Superscript letters (^a,b,c^) indicate subgroups of non-AA treatments that were not significantly different based on post hoc pairwise comparison (Bonferroni test, *p* > 0.05); ^3^ Two-way RM ANOVA (time factor).

## Data Availability

The original data on growth (OD, DW) and intra- vs. extracellular MC/AP peptides have been submitted to the BioStudies database [57] under the Accession numbers S-BSST1672 (DOI 10.6019/S-BSST1672) and S-BSST1681 (DOI 10.6019/S-BSST1681), along with the previous article [23]. The original fluorescent signal data related to this article (A488 signal intensity from high-resolution microscopy, BL1-A original data from FCM) have been deposited under BioStudies Accession numbers S-BSST1898 (DOI 10.6019/S-BSST1898) and S-BSST1906 (DOI 10.6019/S-BSST1906). All corresponding original (deconvolved) image files have been submitted to BioStudies under the Accession number S-BIAD1325 (DOI 10.6019/S-BIAD1325) (https://www.ebi.ac.uk/biostudies, accessed on 28 May 2025).

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
