# Peer review of "Time-Resolved Visualization of Cyanotoxin Synthesis via Labeling by the Click Reaction in the Bloom-Forming Cyanobacteria Microcystis aeruginosa and Planktothrix agardhii"

_toxins, 2025, doi:10.3390/toxins17060278_

Round 1
Reviewer 1 Report
Comments and Suggestions for Authors
Abstract
Line 8. There needs to be a comma between study and time-lapse.
Introduction
Line 32. How does this risk human development?
Results
Tables 1, 2 , 3. What do the letters a, b, and c indicate? It is briefly mentioned in the caption, but please clarify. Please indicate which p-values are significant.
Figures 1, 2, 3, and 4. Please provide better clarity to indicate difference between these figures in the caption. Potential solution could be to bold the section of the caption that is different than the rest.
Figures 5, 6. Please provide better clarity to indicate difference between these figures in the caption. Potential solution could be to bold the section of the caption that is different than the rest. Also, fig 5 contains labeled y-axes in all panels, while fig 6 is only a and b panels; please make this consistent.
Figure 7, 8. It is odd to visual percent on the x-axis as a logarithm. The scale can remain in log, but please change to 1, 10, and 100 for proper visualization.
Discussion
Line 330. I am unsure what is meant by “outermost”.
Materials and Methods
Figure 9. What do the numbers 1, 2, 3 indicate?
Author Response
Comments and Suggestions for Authors
Abstract
Line 8. There needs to be a comma between study and time-lapse. Response: done
Introduction
Line 32. How does this risk human development? the phrase “thus risking human development.” Response: has been deleted
Results
Tables 1, 2 , 3. What do the letters a, b, and c indicate? It is briefly mentioned in the caption, but please clarify. Please indicate which p-values are significant.
Response: In the footnote it is now stated that superscripts indicate subgroups not significantly different in pairwise (post hoc) comparison and overall significant p-values for both factors (grouping and time) are now indicated using superscript stars
Figures 1, 2, 3, and 4. Please provide better clarity to indicate difference between these figures in the caption. Potential solution could be to bold the section of the caption that is different than the rest.
Response: The figure legends Fig. 2-5 have been concised to better indicate the differences. The sentence “Control cells were grown in the absence of non-AAs and processed under identical conditions.” is now mentioned only in Fig.2 legend once.
Figures 5, 6. Please provide better clarity to indicate difference between these figures in the caption. Potential solution could be to bold the section of the caption that is different than the rest. Also, fig 5 contains labeled y-axes in all panels, while fig 6 is only a and b panels; please make this consistent.
Response: The figure legends Fig. 6-7 have been concised as well. Fig.6 labeled y-axes in all panels have been removed in order to indicate that the scale is the same.
Figure 7, 8. It is odd to visual percent on the x-axis as a logarithm. The scale can remain in log, but please change to 1, 10, and 100 for proper visualization.
Response: Fig.8,9 labeled x-axes have been changed from log10 numbers to natural numbers 1,10, 100.
Discussion
Line 330. I am unsure what is meant by “outermost”.
Response: Has been corrected to “utmost importance”
Materials and Methods
Figure 9. What do the numbers 1, 2, 3 indicate?
Response: Figure 10 has been shortened to represent the experimental design while the workflow is now presented using a roadmap diagram in the introduction Fig. 1 (as suggested by R4), thus sentences numbered 1,2,3, have been removed

Reviewer 2 Report
Comments and Suggestions for Authors
Dear authors,
I have carefully reviewed your manuscript highlighting on time resolved visualization of cyanotoxin synthesis via La-2 labelling of amino acids feed by click chemistry approach using bloom-forming cyanobacteria Mi-3 crocystis aeruginosa and Planktothrix agardhii as model organisms.
Results:
The study brings interesting HR microscopy qualitative data of A488 signal generation in cyanobacteria strains during time-lapse build up experiments. While the data is interesting the presentation from tabulated forms to graphical forms is suggested to make the observations more visual. In this regard the following suggestions are presented in results section
L131-133. Table 1. Convert this data to graphical form.
L234-2366. Table 2. Convert this data to graphical form.
L274-276. Table 3. Convert this data to graphical form.
L278-287. Figure 7 and L316-324. Figure 8
Authors mention about LC-MS was used to determine the percentage of clickable MC/AP in total MC/AP. It would be very informative for the readers to visualize this quantitative data in the form of a bar graph separately from these figures in supplement.
Discussion:
L326-369. Sensitivity of chemo selective labeling via Alexa Fluor 488 fluorophore
Authors mention about that flow cytometer is more sensitive as compared to microscopy. It would be very interesting to mention limits of detection of each technique.
Conclusions: Ok
Materials and Methods:
LC-MS analysis section is missing. Please add this section.
L485-507. 5.1. Study organisms and experimental growth conditions. Ok
L515-521. 5.2. Cell harvesting and fixation. Ok
L623-543. 5.3. Chemo-selective labeling of MC/AP synthesis via click reaction. Ok
L544-560. 5.4. Recording of signal labeling in the microscope. Ok
L562-570. 5.5. Calculation of co-localization indices. ok
L598-657. Supplementary data. Ok
References:
Ok
Author Response
L131-133. Table 1. Convert this data to graphical form.
Response: The graphical form of the data in Table 1 is shown in Supplementary material Fig. S1+S2. Since the Table also includes detailed statistical testing we refer to graphical data are shown in Figure S1, S2 in the legend.
L234-2366. Table 2. Convert this data to graphical form.
Response: The graphical form of the data in Table 2 is shown in Fig.6+7 and Fig.S5
Since the Table also includes detailed statistical testing we refer to graphical data are shown in Fig.6+7 and Fig.S5 in the legend.
L274-276. Table 3. Convert this data to graphical form.
Response: The graphical form of the data in Table 3 is shown in Supplementary material Fig. S6+S7.
L278-287. Figure 7 and L316-324. Figure 8
Authors mention about LC-MS was used to determine the percentage of clickable MC/AP in total MC/AP. It would be very informative for the readers to visualize this quantitative data in the form of a bar graph separately from these figures in supplement.
Response: The quantitative data on clickable cyanopeptide synthesis as recorded via LC-MS from aliquots obtained during the same experiments have been reported in the previous publication [23]:
Kurmayer, R. & R. Morón Asensio, 2024. Real-Time Observation of Clickable Cyanotoxin Synthesis in Bloom-Forming Cyanobacteria Microcystis aeruginosa and Planktothrix agardhii. Toxins 16(12):526. https://doi.org/10.3390/toxins16120526
Explanation why it is two articles:
Originally, we wanted to report the whole experiment into one article but realized that it became too much. Thus we decided to separate and report on chemical analysis in [23] and on the relationship between clickable cyanotoxin content and labeling by click reaction in this article.
For this study LC-MS measurements have been included in the Supplement Table S4+S5 (since R4 also asked for including the measurements for each time point).
The original data on growth (OD, DW) and intra- vs. extracellular MC/AP peptides have been submitted to the BioStudies database [57] under the Accession number S-BSST1672 (DOI 10.6019/S-BSST1672) and S-BSST1681 (DOI 10.6019/S-BSST1681) along with the previous article [23].
Discussion:
L326-369. Sensitivity of chemo selective labeling via Alexa Fluor 488 fluorophore
Authors mention about that flow cytometer is more sensitive as compared to microscopy. It would be very interesting to mention limits of detection of each technique.
Response: L378-390: in the discussion section the comparison in sensitivity has been elaborated:
“In FCM the rectangular gating tool allows to discriminate relatively low numbers of particles as compared to controls, supporting the early discrimination of labeled from non-labeled subpopulations. Under the specified conditions, during time-lapse build up for M. aeruginosa only 27-149 “false” positive particles and for P. agardhii 0-33 “false” positive particles were differentiated resulting in 0.04-0.69% and 0-1.75 % of the total population, respectively (n = 18). For FCM the observed noise would imply a limit of detection around 1%. In contrast the natural A488 signal intensities per cell volume recorded from controls showed higher varia-tion, i.e. during time-lapse build up for M. aeruginosa the A488 intensity in controls varied from 50 – 150% and for P. agardhii it varied from 75-118% (n = 18). Thus for A488 labeling the potential limit of detection would imply a 1.5-fold exceed of natural A488 intensity in M. aeruginosa and a 1.2-fold exceed of natural A488 intensity in P. agardhii.”
Conclusions: Ok
Materials and Methods:
LC-MS analysis section is missing. Please add this section.
Response: Under section 5.2. Cell harvesting and fixation the following has been added:
“In parallel cells were filtered on glass fiber filters by vacuum filtration and extracted for natural and clickable peptides as described in the previous article [23]. The quantitative data on clickable cyanopeptide synthesis as recorded via LC-MS from aliquots obtained during the same experiments have been reported in the previous publication [23] and were used to ex-plore the relationship between clickable peptide content and labeling by click reaction in this article.”
Under section 5.8. chemical analytical analysis
The differentiated natural/clickable MCs and APs are described.

Reviewer 3 Report
Comments and Suggestions for Authors
The manuscript presents a novel approach for visualizing the in vivo synthesis dynamics of cyanotoxins (microcystins and anabaenopeptins) in two ecologically relevant cyanobacteria species using a combination of precursor-directed biosynthesis with non-natural amino acids containing clickable handles and subsequent fluorescent labeling via click chemistry. This approach, combined with time-lapse microscopy and flow cytometry, offers a significant advancement in understanding the spatial and temporal regulation of cyanotoxin production at the cellular and population levels. Therefore, the manuscript has the potential to be suitable for publication.
I have highlighted some of the issues that need to be addressed before its consideration for publication. These are given below:
Although the abstract provides a good overview, it needs to be slightly more concise. For example, the repetition of “clickable MCs or Aps” needs to be streamlined. Consider rephrasing sentences for better flow and impact.
Even though M. aeruginosa and P. agardhii are common bloom-forming species, a stronger justification for selecting these two specific species, beyond their bloom-forming nature and differing autofluorescence, would be beneficial. Are there known differences in their NRPS machinery or regulation that make this comparative study particularly insightful?
The lack of A488 signal in P. agardhii fed with Phe-Az despite evidence of incorporation is a key finding that warrants further investigation. Although the authors propose in vitro click chemistry as a next step, the discussion should be expanded to consider other potential reasons for this observation within the cellular context (e.g., altered accessibility of the azide group, different localization hindering the reaction).
The consistent observation of heterogeneous A488 signal distribution is intriguing and the promise of a forthcoming article detailing this is noted. However, a slightly more in-depth discussion of potential hypotheses for this heterogeneity (e.g., localized synthesis sites, differential toxin accumulation within the cell) within this manuscript would add value.
Whereas the log transformation of data to improve the coefficient of determination (R²) is mentioned, a brief justification for this transformation (e.g., non-normal distribution of the original data, variance stabilization) would be helpful for transparency.
The statistical methods used (RM ANOVA, Bonferroni t-test, linear regression) appear appropriate. However, ensuring that all statistical assumptions were met and providing details on the statistical software used would strengthen the rigor.
The discussion on potential applications in gene function studies is interesting. Expanding on specific examples of how this technology could be used to investigate the role of individual NRPS domains or regulatory elements would enhance the impact of the work.
The English language in the manuscript is generally of good quality and easily understandable. However, there are some instances where minor grammatical errors and problematic and poor phrasing occur.
Despite the minor drawbacks and suggestions for improvement, the manuscript presents a valuable and innovative methodology for studying cyanotoxin synthesis. The combination of chemical labeling and advanced imaging techniques provides a powerful tool for investigating the spatio-temporal dynamics of these important secondary metabolites. Addressing the suggested points, particularly clarifying the Phe-Az results in P. agardhii will further enhance the manuscript and its suitability for publication.
Comments on the Quality of English LanguageThe English language in the manuscript is generally of good quality and easily understandable. However, there are some instances where minor grammatical errors and problematic and poor phrasing occur.
Author Response
I have highlighted some of the issues that need to be addressed before its consideration for publication. These are given below:
Although the abstract provides a good overview, it needs to be slightly more concise. For example, the repetition of “clickable MCs or Aps” needs to be streamlined. Consider rephrasing sentences for better flow and impact.
Response: abstract has been rephrased to read more specific and for better reading flow
Even though M. aeruginosa and P. agardhii are common bloom-forming species, a stronger justification for selecting these two specific species, beyond their bloom-forming nature and differing autofluorescence, would be beneficial. Are there known differences in their NRPS machinery or regulation that make this comparative study particularly insightful?
Response: l.46-57: A paragraph on more details of NRPS as well as structural diversity in metabolite synthesis has been included.
The lack of A488 signal in P. agardhii fed with Phe-Az despite evidence of incorporation is a key finding that warrants further investigation. Although the authors propose in vitro click chemistry as a next step, the discussion should be expanded to consider other potential reasons for this observation within the cellular context (e.g., altered accessibility of the azide group, different localization hindering the reaction).
Response: L434-447: this part has been re-written, Firstly it is stated that the possible reasons for azide-based toxicity have been discussed [23] and the exceptional reactivity of the azide group is known (Schock & Bräse 2020). Since autofluorescence also decreased in both cyanobacteria (Figure 7), it is argued that one possible reason for reduced AF might be (uncontrolled) intracellular decomposition of Phe-Az resulting in reactive nitrenes causing the overall toxic side effects. Thus, though Phe-Az is incorporated into MC/AP to some extent most of the non-incorporated Phe-Az in the cell might have resulted in nitrenes counteracting the expected A488 fluorescence signal. Obviously, M. aeruginosa is much less sensitive when compared with P. agardhii (M. aeruginosa also had high cellular contents of Phe-Az, Suppl Fig S9 in [23]).
The consistent observation of heterogeneous A488 signal distribution is intriguing and the promise of a forthcoming article detailing this is noted. However, a slightly more in-depth discussion of potential hypotheses for this heterogeneity (e.g., localized synthesis sites, differential toxin accumulation within the cell) within this manuscript would add value.
Response: L456-466: the two potential hypotheses (i.e. storage or accumulation vs biosynthesis have been now more directly addressed and will be tested more explicitly in a forthcoming article.
Whereas the log transformation of data to improve the coefficient of determination (R²) is mentioned, a brief justification for this transformation (e.g., non-normal distribution of the original data, variance stabilization) would be helpful for transparency.
Response: L645-654: It is now stated: “For linear regression analysis the data on clickable MC/AP in total MC/AP content and A488 signal intensity as recorded by both microscopy and FCM were log(x+1) transformed to make the data linear.”
Since assumptions for normality and equal variance were not met Spearman Rank Order Correlation was also tested and confirmed positive relationships in each case (e.g. Figure 8+9, Figure S8+S13).
The statistical methods used (RM ANOVA, Bonferroni t-test, linear regression) appear appropriate. However, ensuring that all statistical assumptions were met and providing details on the statistical software used would strengthen the rigor.
Response: L640-644: For RM ANOVA it is now stated: “Assumptions on normality (Shapiro-Wilk) were not met in part for A488 intensity and not met for A488/AF ratio, while equality of variance (Brown-Forsythe) was always passed (p < 0.05).”
L654: “All tests were performed using Sigma plot for Windows version 14.0.”
The discussion on potential applications in gene function studies is interesting. Expanding on specific examples of how this technology could be used to investigate the role of individual NRPS domains or regulatory elements would enhance the impact of the work.
Response: L511-521: one more paragraph to investigate possible regulatory elements has been added.
The English language in the manuscript is generally of good quality and easily understandable. However, there are some instances where minor grammatical errors and problematic and poor phrasing occur.
Response: After this revision, we would be happy to order a final proofreading using a commercial service , e.g. proof-reading-service.com .
Despite the minor drawbacks and suggestions for improvement, the manuscript presents a valuable and innovative methodology for studying cyanotoxin synthesis. The combination of chemical labeling and advanced imaging techniques provides a powerful tool for investigating the spatio-temporal dynamics of these important secondary metabolites. Addressing the suggested points, particularly clarifying the Phe-Az results in P. agardhii will further enhance the manuscript and its suitability for publication.
Response: Basically, I think it has to do with ongoing generation of reactive nitrenes through intracellular Phe-Az counteracting click-generated fluorescence (since autofluorescence also has been dramatically reduced, though M. aeruginosa appeared less sensitive).

Reviewer 4 Report
Comments and Suggestions for Authors
Time-Resolved Visualization of Cyanotoxin Synthesis via Labeling by Click Reaction in Bloom-Forming Cyanobacteria Microcystis aeruginosa and Planktothrix agardhii
(CRITICAL REVIEW)
Comments about the abstract:
- Lines 5-7:
Consider revising the sentence:
“It has been shown that promiscuous adenylation (A) domains in non-ribosomal peptide synthesis (NRPS) allow the incorporation of clickable non-natural amino acids (non-AAs) into their peptide products, i.e., microcystins (MCs) or anabaenopeptins (APs).”
To:
“Promiscuous adenylation (A) domains in non-ribosomal peptide synthesis (NRPS) allow the incorporation of clickable non-natural amino acids (non-AAs) into their peptide products, i.e., microcystins (MCs) or anabaenopeptins (APs).”
This change is recommended because, as currently written, the phrase "It has been shown..." implies that you are referencing previous work. If you intend to do so, a proper citation should be provided
- Please present the data and findings in the abstract. The abstract of the manuscript is empirical rather than data-based.
- Lines 6 and 9: Please use the complete form of non-AAs once.
Lines 8-9: “In a second step, click chemistry is used for visualization of chemically modified cyano-peptides.” It should be “In a second step, click chemistry is used to visualise chemically modified cyano-peptides.” It is not clear what the second step means and what the first step is. Please connect the abstract theme properly.
- Lines 9-12: “ In this study time-lapse experiments have been performed using pulsed feeding of non-natural amino acids (non-AAs) in order to observe the build-up or decline of azide- or alkyne-modified peptides, i.e. clickable MCs or APs using chemo selective labeling via Alexa Fluor 488 (A488) fluorophore for visualizing MC/AP. “ this sentence is too long and dense please revise it.
- Lines 12-13: “ In particular the results on A488 signal intensities were compared directly to the clickable AP/MC content as reported earlier.” What do the authors mean here reported earlier, there should be a proper and precise description of this.
- The study title is “Time-Resolved Visualization of Cyanotoxin Synthesis via Labelling by Click Reaction in Bloom-Forming Cyanobacteria Microcystis aeruginosa and Planktothrix agardhii. However, I do not see the mention of Cyanotoxin Synthesis, Microcystis aeruginosa, or Planktothrix agardhii in the abstract. Consider revising the title or mentioning them in the abstract.
- Keywords are too many. Please reduce them.
- Line 11: Chemoselective instead chemo selective.
- Please remove key contribution from the abstract.
- Write the complete form of cHAB
- Lines 31-34: “Cyanotoxin production is a global problem as it deteriorates freshwater water quality thus risking human development.” I think here term “human development” is not appropriate.
- Lines 42-43: “ the thiotemplate mechanism, known as non-ribosomal peptide synthesis (NRPS). The complete form of NRPS has been used earlier in the abstract. It is suggested to use the abbreviation throughout the manuscript.
- It is suggested that the complete form of the abbreviation be given first, then the abbreviation should be used in the manuscript.
- In the introduction, please give details about non-ribosomal peptide synthesis (NRPS) and click chemistry.
- Please introduce the time-lapse experiments in the introduction section.
- In the Table 1, please add a column present % increase or decrease build-up.
- Please provide a beautiful diagram presenting the roadmap of the study for new users.
- The time-lapse signal build-up and decline experiments span over 168 hours. It would have been if data for all points were presented in Table 1.
- In the result section, “Quantification of A488 signal intensity and autofluorescence via high-resolution microscopy in cyanobacteria strains during time-lapse build-up and decline experiments was measured upto 7 days. It would be better if, within the result section, % or fold increase or decrease signal intensity is mentioned. I know data has been presented in the graph, but the same should have been mentioned in the result. This should be done for all quantification experiments.
- Lines 328-330: “While methods to reduce AF interference have been developed continuously [23,24] the choice of the appropriate fluorophore A488 is of outermost importance [25].” Could not get this statement, and I think it should be of “utmost importance”. There are several other mistakes similar to this, such as difficulty understanding the message of the author. Therefore, it is recommended that a colleague proofread the manuscript.
- Lines 330-332: “It could be shown that A488 labeling signal reacted sensitively, which is in correspondence to the early synthesis of click- 331 able MC/AP as recorded from peptide extracts reported earlier.” consider revising “ The A488 labeling signal responded sensitively, which corresponds to the early synthesis of clickable MC/AP as observed in peptide extracts reported earlier.”
- Lines 339-341: “Both aeruginosa and P. agardhii contained low amounts of clickable MC/AP in total MC or AP as early as at T0, i.e. for Phe-Az 5 ± 1 (SE)% and for Prop-Tyr 4 ± 1% for M. aeruginosa as well as 3 ± 2% for Prop-Lys in P. agardhii, i.e. Figs. 1,2 in [16]. What indicates “in” and is it appropriate? Please check. Please explain why Both M. aeruginosa and P. agardhii contained low amounts of clickable MC/AP in total MC or AP?
- Lines 400-401: “The possible reasons for azide-based toxicity have been discussed [16].” It is better explain the reason in the manuscript as well.
- In the results, only AP has been discussed. Anabaenopeptins are a diverse group of cyclic hexapeptide protease inhibitors produced by cyanobacteria, with notable examples including Anabaenopeptin A, B, F, and 915, among others. It is not clear which AP group has been produced.
- Micropeptins are cyclic peptides produced by certain cyanobacteria. They are classified into types based on their structures and biological activities, with some examples including micropeptin T-20 and micropeptin SF909. As a result, only MC build-up or decline has been shown without the specification.
- The light intensity is written as:"50 .m-2 .s-1" This appears to be an error. The correct unit should be µmol photons m⁻² s⁻¹ (standard for PAR).
- The phrase "Fixed cells and stored in 1x PBS at 4°C in the dark the previous day" is unclear. Reword it to specify whether cells were fixed the day before and stored until use, or if they were stored and fixed the same day. Please specify why this adjustment was made. Was it to reduce the background signal? The sentence "except that emission spectral range in channel 0 was narrowed down from 504-580 nm to 500-550 nm" could be clearer.
Additional rectification
All figures and tables are extending out of paragraphs.
Page No. 18
- Non- AA should be Non standard AA
- cyanobacterium Synechococcus 7942 (the genus Cyanobacterium is not italicized)
- (see [35] for review) looks unusual
Page No. 17
- (see [16] for review) looks unusual
- HPLC-MSn (superscript n is not defined)
Page No. 16
- Non- AA should be Non standard AA
- Moschny et al. [28] look unusual because the reference is used in the text and cited.
Page No. 09
(Figure 5a, 192 Figure S3). Didnt find Figure S3

Many issues with grammar, writing style, and the use of scientific terms make reading and understanding the manuscript difficult.
Author Response
Comments about the abstract:
- Lines 5-7:
Consider revising the sentence:
“It has been shown that promiscuous adenylation (A) domains in non-ribosomal peptide synthesis (NRPS) allow the incorporation of clickable non-natural amino acids (non-AAs) into their peptide products, i.e., microcystins (MCs) or anabaenopeptins (APs).”
To:
“Promiscuous adenylation (A) domains in non-ribosomal peptide synthesis (NRPS) allow the incorporation of clickable non-natural amino acids (non-AAs) into their peptide products, i.e., microcystins (MCs) or anabaenopeptins (APs).”
This change is recommended because, as currently written, the phrase "It has been shown..." implies that you are referencing previous work. If you intend to do so, a proper citation should be provided
Response: done
- Please present the data and findings in the abstract. The abstract of the manuscript is empirical rather than data-based.
- Lines 6 and 9: Please use the complete form of non-AAs once.
Response: done, the abstract has been rewritten and more results have been included
Lines 8-9: “In a second step, click chemistry is used for visualization of chemically modified cyano-peptides.” It should be “In a second step, click chemistry is used to visualise chemically modified cyano-peptides.” It is not clear what the second step means and what the first step is. Please connect the abstract theme properly.
Response: Has been corrected
- Lines 9-12: “ In this study time-lapse experiments have been performed using pulsed feeding of non-natural amino acids (non-AAs) in order to observe the build-up or decline of azide- or alkyne-modified peptides, i.e. clickable MCs or APs using chemo selective labeling via Alexa Fluor 488 (A488) fluorophore for visualizing MC/AP. “ this sentence is too long and dense please revise it.
Response: This sentence has been shortened
- Lines 12-13: “ In particular the results on A488 signal intensities were compared directly to the clickable AP/MC content as reported earlier.” What do the authors mean here reported earlier, there should be a proper and precise description of this.
Response: It is now stated: “In particular the resulting Alexa Fluorophore 488 signal intensities were linearly related to the clickable AP/MC content as recorded by chemical analytical technique.”
- The study title is “Time-Resolved Visualization of Cyanotoxin Synthesis via Labelling by Click Reaction in Bloom-Forming Cyanobacteria Microcystis aeruginosa and Planktothrix agardhii. However, I do not see the mention of Cyanotoxin Synthesis, Microcystis aeruginosa, or Planktothrix agardhii in the abstract. Consider revising the title or mentioning them in the abstract.
Response: has been now included
- Keywords are too many. Please reduce them.
Response: number of keywords has been reduced
- Line 11: Chemoselective instead chemo selective.
Response: Chemoselective has been corrected throughout
- Please remove key contribution from the abstract.
Response: Sentence for key contribution has been removed from the abstract
- Write the complete form of cHAB
Response: done
- Lines 31-34: “Cyanotoxin production is a global problem as it deteriorates freshwater water quality thus risking human development.” I think here term “human development” is not appropriate.
Response: has been removed
- Lines 42-43: “ the thiotemplate mechanism, known as non-ribosomal peptide synthesis (NRPS). The complete form of NRPS has been used earlier in the abstract. It is suggested to use the abbreviation throughout the manuscript.
Response: NRPS has been used throughout
- It is suggested that the complete form of the abbreviation be given first, then the abbreviation should be used in the manuscript.
Response: has been followed
- In the introduction, please give details about non-ribosomal peptide synthesis (NRPS) and click chemistry.
Response: L42-L45: more details on NRPS have been added (R3), and the observed evolved changes to increase structural diversity in NRPS have been described (L46-57). One general sentence on click chemistry has been added (L79-84)
- Please introduce the time-lapse experiments in the introduction section.
Response: L107-112: 2 sentences have been added.
- In the Table 1, please add a column present % increase or decrease build-up.
Response: In the table 2 mean ±SE fold changes (as compared to control) have been added for A488 intensity and autofluorescence (in parentheses)
- Please provide a beautiful diagram presenting the roadmap of the study for new users.
Response: a workflow diagram has been added (new Figure 1)
- The time-lapse signal build-up and decline experiments span over 168 hours. It would have been if data for all points were presented in Table 1.
Response: More detailed tables on the measurements for each time (Suppl Table S1-5) have been included in the Supplementary
- In the result section, “Quantification of A488 signal intensity and autofluorescence via high-resolution microscopy in cyanobacteria strains during time-lapse build-up and decline experiments was measured upto 7 days. It would be better if, within the result section, % or fold increase or decrease signal intensity is mentioned. I know data has been presented in the graph, but the same should have been mentioned in the result. This should be done for all quantification experiments.
Response: x-fold increase or decrease has been included in the text
- Lines 328-330: “While methods to reduce AF interference have been developed continuously [23,24] the choice of the appropriate fluorophore A488 is of outermost importance [25].” Could not get this statement, and I think it should be of “utmost importance”. There are several other mistakes similar to this, such as difficulty understanding the message of the author. Therefore, it is recommended that a colleague proofread the manuscript.
Response: The sentence has been modified. After this revision round, I would order a final proofreading using a commercial service , e.g. proof-reading-service.com .
- Lines 330-332: “It could be shown that A488 labeling signal reacted sensitively, which is in correspondence to the early synthesis of click- 331 able MC/AP as recorded from peptide extracts reported earlier.” consider revising “ The A488 labeling signal responded sensitively, which corresponds to the early synthesis of clickable MC/AP as observed in peptide extracts reported earlier.”
Response: has been revised
- Lines 339-341: “Both aeruginosa and P. agardhii contained low amounts of clickable MC/AP in total MC or AP as early as at T0, i.e. for Phe-Az 5 ± 1 (SE)% and for Prop-Tyr 4 ± 1% for M. aeruginosa as well as 3 ± 2% for Prop-Lys in P. agardhii, i.e. Figs. 1,2 in [16]. What indicates “in” and is it appropriate? Please check. Please explain why Both M. aeruginosa and P. agardhii contained low amounts of clickable MC/AP in total MC or AP?
Response: “in total MC or AP” has been removed.
L369-371: The early incorporation of non-AA at T0 can be explained by the manipulation time (approx. 1 h) including centrifugation and washing in PBS until fixing in 2% PFA.
- Lines 400-401: “The possible reasons for azide-based toxicity have been discussed [16].” It is better explain the reason in the manuscript as well.
Response: L434-447: this part has been re-written, (see also R3)
In the results, only AP has been discussed. Anabaenopeptins are a diverse group of cyclic hexapeptide protease inhibitors produced by cyanobacteria, with notable examples including Anabaenopeptin A, B, F, and 915, among others. It is not clear which AP group has been produced.
Response: L657-665: a paragraph on which natural and clickable MC/Aps have been quantified has been added (Section 5.8. Chemical analytical analysis)
- Micropeptins are cyclic peptides produced by certain cyanobacteria. They are classified into types based on their structures and biological activities, with some examples including micropeptin T-20 and micropeptin SF909. As a result, only MC build-up or decline has been shown without the specification.
Response: Micropeptins would imply cyanopeptolins which so far have not been tested directly. However, for P. agardhii CYA126/8 a cyanopeptolin synthesis gene inactivation mutant has been tested which did not reveal incorporation of either non-AA (Moron et al. 2021, Microorganisms, https://www.mdpi.com/2076-2607/9/8/1578 )
- The light intensity is written as:"50 .m-2 .s-1" This appears to be an error. The correct unit should be µmol photons m⁻² s⁻¹ (standard for PAR).
Response: has been corrected
- The phrase "Fixed cells and stored in 1x PBS at 4°C in the dark the previous day" is unclear. Reword it to specify whether cells were fixed the day before and stored until use, or if they were stored and fixed the same day. Please specify why this adjustment was made. Was it to reduce the background signal? The sentence "except that emission spectral range in channel 0 was narrowed down from 504-580 nm to 500-550 nm" could be clearer.
Response: it is now stated: “Cells were fixed the day before and stored in 1x PBS at 4°C in the dark until use …”
this type of workflow has been established since the beginning, as harvesting cells and cell fixation takes several hours (along with vacuum filtration for peptide extraction in parallel)
The emission spectral range in channel 0 (A488) was narrowed down from 504-580 nm to 500-550 nm to enable another fluorophore to allow for dual labeling (A555). (unpublished)
Additional rectification
All figures and tables are extending out of paragraphs.
All tables and graphs have been adjusted that they are not extending on the right side anymore.
Page No. 18
- Non- AA should be Non standard AA
we used “non-AA” as abbreviation for non-natural amino acid also in the related article describing clickable MC/AP synthesis as revealed by chemical-analytical technique [23], Kurmayer & Moron 2024, Toxins, https://www.mdpi.com/2072-6651/16/12/526
- cyanobacterium Synechococcus 7942 (the genus Cyanobacterium is not italicized)
- (see [35] for review) looks unusual
Response: corrected
Page No. 17
- (see [16] for review) looks unusual
- HPLC-MSn (superscript n is not defined)
Response: corrected
Page No. 16
- Non- AA should be Non standard AA
- Moschny et al. [28] look unusual because the reference is used in the text and cited.
Response: corrected
Page No. 09
(Figure 5a, 192 Figure S3). Didnt find Figure S3
Response: Figure S1-S13 are in the supplementary, as well as new Tables S1-S5 (with measurements for each time point have been added)

Round 2
Reviewer 3 Report
Comments and Suggestions for Authors
The critical review of the revised version of the manuscript showed that all the queries made in the first manuscript have been dealt with diligently and the manuscript has been considerably improved. There is substantial improvement in the revised manuscript both technically and linguistically.